# The chromatin network helps prevent cancer-associated mutagenesis at transcription-replication conflicts

Aleix Bayona-Feliu [1,2,3] ✉, Emilia Herrera-Moyano [1,2], Nibal Badra-Fajardo[1], Iván Galván-Femenía[3], María Eugenia Soler-Oliva [1,2] & Andrés Aguilera [1,2] ✉

Genome instability is a feature of cancer cells, transcription being an important source of DNA damage. This is in large part associated with R-loops, which hamper replication, especially at head-on transcription-replication conflicts (TRCs). Here we show that TRCs trigger a DNA Damage Response (DDR) involving the chromatin network to prevent genome instability. Depletion of the key chromatin factors INO80, SMARCA5 and MTA2 results in TRCs, fork stalling and R-loop-mediated DNA damage which mostly accumulates at S/G2, while histone H3 Ser10 phosphorylation, a mark of chromatin compaction, is enriched at TRCs. Strikingly, TRC regions show increased mutagenesis in cancer cells with signatures of homologous recombination deficiency, transcription-coupled nucleotide excision repair (TC-NER) and of the AID/APOBEC cytidine deaminases, being predominant at head-on collisions. Thus, our results support that the chromatin network prevents R-loops and TRCs from genomic instability and mutagenic signatures frequently associated with cancer.

Cancer is a multifactorial disease triggered by the dysfunction of several cellular processes that fuel uncontrolled cell proliferation. Genome instability (GIN) is a hallmark of cancer cells and one of the main causes of cell transformation and cancer evolution[1]. Unstable genomes favor mutagenesis of tumor suppressors and oncogenes thus promoting malignant cell transformation and facilitating tumor adaptation to a wide range of scenarios, including therapeutic treatments. Remarkably, chromatin-regulating enzymes are emerging as key factors in maintaining genome integrity[2], and fine-tuning of chromatin structure and function has been linked to prevent transcription-associated DNA damage thus ensuring genome fidelity. Consistently, chromatin-modulating activities are frequently found altered in tumors[3].

The transcription machinery may pose a roadblock to replication fork progression, promoting replication stress and DNA damage[4] that can potentially be enhanced by the occurrence of non-B DNA structures such as R-loops, a three-stranded nucleic acid structure composed by a DNA–RNA hybrid and a displaced single strand DNA (ssDNA) that form co-transcriptionally. Although R-loops have physiological roles in particular processes such as class switch recombination, they may pose a threat to genome integrity. Consequently, cells possess several mechanisms to counteract the pathological accumulation of R-loops, either by preventing their formation via RNA binding or processing factors, such as the THO complex, SRSF1 and TDP43 among others[5–10], or by resolving factors like DNA:RNA helicases such as SETX, UAP56/DDX39b or DDX5[11–15] and RNases H, such as RNH1 or RNH2[16]. In addition, R-loops may indirectly be resolved by the action of DDR factors, such as BRCA1, BRCA2, FANCD2 or ATR, among others[17–21].

The Encyclopedia of DNA Elements (ENCODE) project[22,23] allows access to a large number of genome-wide data for comparative studies, in particular from K562 cells, used mostly as a standard for

[1]Centro Andaluz de Biología Molecular y Medicina Regenerativa CABIMER, Universidad de Sevilla-CSIC-Universidad Pablo de Olavide, 41092 Seville, Spain. [2]Departamento de Genética, Facultad de Biología, Universidad de Sevilla, 41012 Seville, Spain. [3]Institute for Research in Biomedicine (IRB Barcelona), The Barcelona Institute of Science and Technology (BIST), Barcelona, Spain. ✉e-mail: aleix.bayona@irbbarcelona.org; aguilo@us.es

genome-wide studies. Furthermore, cancer-associated mutagenesis is also accessible through the Catalogue of Somatic Mutations in Cancer (COSMIC) database[24], enabling identification of mutational signatures associated with a specific etiology. Recently, the availability of DRIPc-seq (DNA:RNA hybrid immunoprecipitation followed by sequencing of the cDNA derived from the RNA moiety of hybrids) and OK-seq data (Okazaki fragment sequencing)[25] in K562 cells allowed us to predict genomic sites prone to TRCs[26].

Here, we show that in addition to the SWI/SNF complex, other chromatin remodeler factors such as INO80, SMARCA5 and MTA2 contribute to prevent R-loop accumulation in cells. Then, we used the ENCODE genome-wide data in K562 cells and COSMIC database to cross ChIP-seq and cancer-associated mutagenesis with our genomic TRC dataset to unveil possible structural and functional features of TRC sites. We show that a wide range of chromatin remodelers, modifiers and epigenetic marks, as well as transcription and DDR factors are enriched at TRC sites, in particular at those occurring in head-on orientation. This correlates with a high incidence of single nucleotide variations (SNVs) and insertions and deletions (indels) mutations at such sites in tumoral cells, as found in the COSMIC database. Strikingly, such mutagenic signatures are found highly enriched at sites where R-loops and head-on TRCs occur preferentially in cells depleted of SWI/SNF. Altogether, these results establish a direct link between the DDR and epigenetic factors at TRCs and their contribution to prevent mutagenesis and genomic instability mediated by DSB repair and translesion synthesis (TLS) associated with fork stalling that is over-represented in cancer cells. This study may open new perspectives in the design and development of new cancer therapies.

## Results

### Enrichment of chromatin, DNA, and RNA metabolism factors at TRCs

Different studies have addressed the contribution of DNA repair factors on DNA–RNA hybrid accumulation and its impact on genome integrity. The results support the view that DNA repair factors regulate DNA–RNA hybrid homeostasis upon fork stalling, DSBs formation and post-replicative repair (PRR)[17–21,27,28]. The relevance of chromatin to this process has been provided by different studies[26,29–32], but it seems yet incomplete considering the high number of factors influencing chromatin function. Indeed, the frequency of mutations in chromatin remodeler and modifier genes is high in cancer, genome instability being a hallmark of tumor cells. Accordingly, the SWI/SNF complex, one of the most frequently altered chromatin complexes in cancer, plays a key role helping solve TRCs[26,30].

Consequently, we did a global search for additional chromatin and functionally interacting factors. Despite the large number of genes demonstrated or proposed to regulate R-loop homeostasis and therefore to impact on TRCs and genome integrity, our knowledge on specific factors associated with TRC sites is still limited. The ENCODE Project provides massive, curated genome-wide data that can be used for this purpose, particularly regarding factors involved in chromatin organization. Thus, we analyzed the abundance of the whole catalog of proteins for which ChIP-seq data are available in ENCODE at R-loop-enriched TRC sites to unveil the factors and molecular processes contributing to preserve genome integrity at TRCs. We first gathered the ENCODE ChIP-seq data available for the K562 cell line and integrated them with the R-loop sites obtained by DRIPc-seq, where replication fork directionality (RFD) is mostly homogenous among cell population (RFD > |0.75|), to ensure accurate and directional TRC analysis as previously shown[26]. As control, we also included FANCD2 ChIP-seq data[33] to the analysis, a standard marker for replication fork stalling[34] previously described to accumulate at these sites[26]. Then, we plotted the coverage of proteins along R-loop-enriched TRC sites according to the RFD and measured protein abundance at TRCs (Fig. 1a). We extended the plots to +/−1 Mb from peaks as the DNA

damage foci highlighted by γH2AX may extend up to 1–2 Mb in mammals and we were analyzing DDR factors.

Next, we performed gene set enrichment analysis (GSEA) to determine whether specific chromatin functional categories could be retrieved either enriched or depleted at the R-loop-rich TRC sites. To do so, we selected a subset of Gene Ontology (GO) functional categories including chromatin, but also transcription, DNA replication and DNA damage GO terms as controls, and we searched for significant enrichments using the GSEA software (UC San Diego and Broad Institute)[35,36]. First, we screened for factors whose abundance was significantly changed at the R-loop-enriched TRC sites in respect of the surrounding region by filtering those factors with Rank Metric Scores (RMS) > | 0.25| at TRCs. Our analysis revealed 209 factors that were enriched and 12 depleted (Fig. 1b). In accordance with published data, several factors previously described to participate in R-loop metabolism including SMARCA4, FANCD2, histone acetylation, BRD4, EP400, YY1 and EWSR1 were found enriched at TRCs[17,18,26,32,37–40]. Consistently, proteins encompassing transcription by RNA polymerase II (RNAPII), as well as factors involved in the DNA replication and DNA Damage Response (DDR) were also found enriched at TRC sites (Fig. 1c, Supplementary Fig. 1a–f, and Supplementary Table 1). Indeed, the FANCD2 Fanconi Anemia factor, the RAD51 recombinase (RAD51), the nibrin (NBN) component of the MRN complex and the RAD21 cohesin component emerged in our analysis as DNA repair factors peaking at TRCs. Strikingly however, canonical DNA replication components were observed dropping significantly at these regions (Fig. 1c, Supplementary Fig. 1e, and Supplementary Table 2), with a significant reduction in replicative helicases as indicated by the helicase GO term. These results are consistent with fork collapse and DNA breakage at TRCs as largely reported[17,18,21,28,41], suggesting that these events are concomitant to disengaging of the replicative MCM helicase followed by the action of DSB repair factors.

In addition to the above-mentioned factors harboring a functional role during DNA and RNA metabolism, an important number of epigenetic factors were also found associated with TRC sites, including chromatin remodelers and modifiers as well as specific epigenetic marks (Fig. 1d, Supplementary Fig. 1d, and Supplementary Table 1). Thus, components of the 4 main chromatin remodeling families (SWI/SNF, ISWI, INO80 and CHD) were enriched at TRC sites. More specifically, these comprised the factors of the SWItch/Sucrose Non Fermentable (SWI/SNF) complex SMARCA4, SMARCC2/BAF170, ARID1B and ARID2, the members of the Inositol 80 (INO80) family EP400 and YY1, the component of the Imitation SWItch (ISWI) complex SMARCA5, and the MTA members of the Nucleosome Remodeling Deacetylase (NuRD) complex, which belongs to the Chromodomain helicase DNA-binding (CHD) remodeling family. In addition, chromatin modifiers mostly involved in histone acetylation and methylation were also overrepresented at TRCs (Fig. 1d, Supplementary Fig. 1d, and Supplementary Table 1). The first class included the HDAC1, HDAC3 and SIN3B histone deacetylase complexes (HDACs), the GATAD2B component of the MeCP1 HDAC complex, as well as the bromo-domain factor BRD4, whereas the second included the PHF8, KDM1A histone lysine demethylase (KDM) and the L3MBTL2 histone methyl-lysine binding protein of the Polycomb group (PcG). Consistently, histones H3 and H4 mono-methylated H3K9me1 and H4K20me1, di-methylated H3K4me2 and H3K79me2, and tri-methylated H3K4me3 and H3K36me3, as well as acetylated H3K9ac and H3K27ac were increased at TRC sites. Moreover, components regulating DNA methylation and higher order chromatin structure such as DNMT1 and TRIP13 or FOXA1, respectively, were also found enriched at TRCs.

Interestingly, similar to the underrepresentation of the replicative MCM helicases, some specific epigenetic marks and factors were significantly reduced at TRCs (Fig. 1c, Supplementary Fig. 1d, and Supplementary Table 2). These included the H3K27me3 epigenetic mark, and the CBX8 chromo-box member of the PcG PRC1-like complex, the

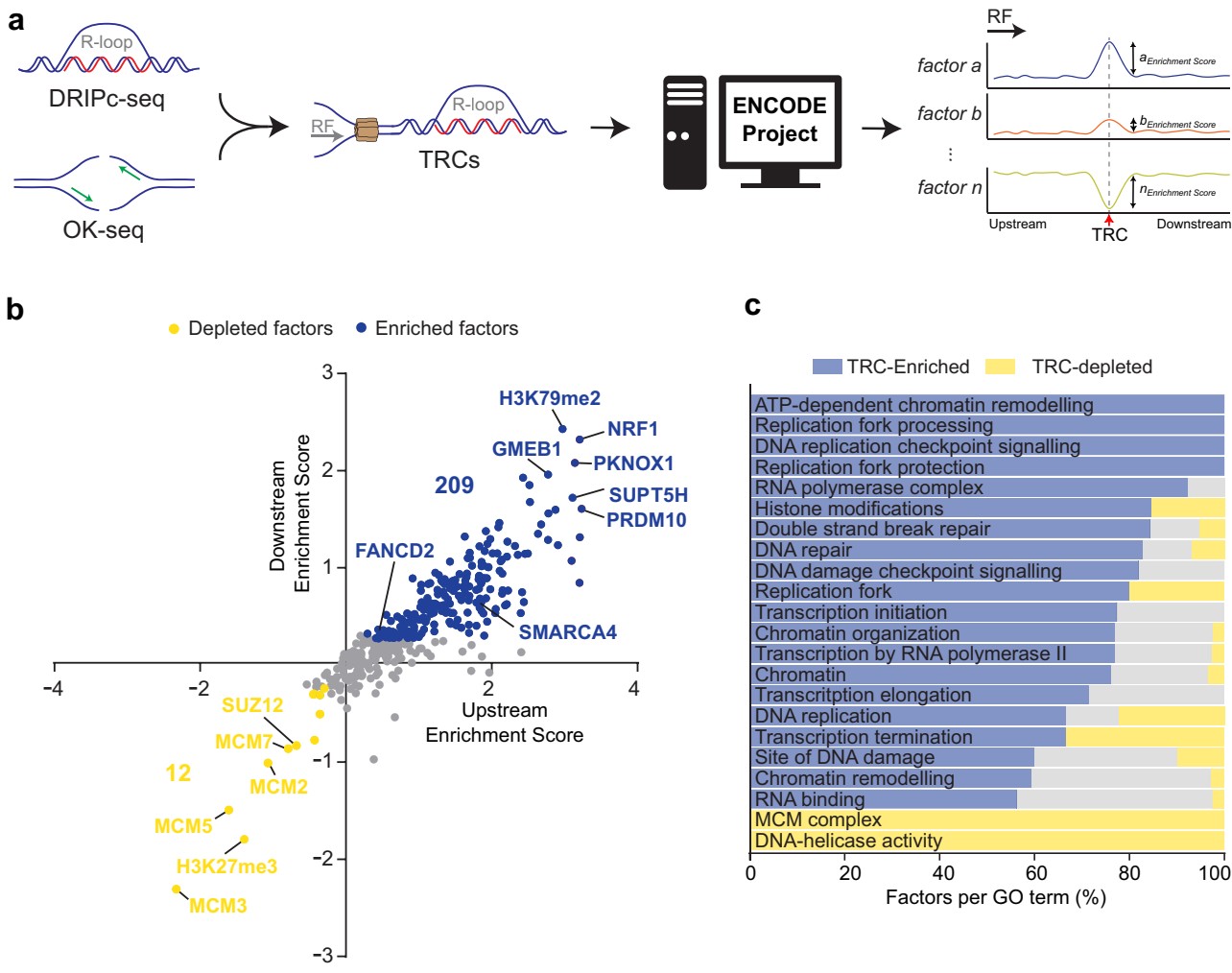

**Fig. 1 | Factors enriched and depleted at transcription-replication conflicts (TRCs). a** Schematic summary of the ChIP-seq data downstream processing for enrichment/depletion analysis at TRCs. **b** Scatter plot indicating the fold change in content for each factor when comparing the TRC site versus its upstream and downstream sequences. **c** Percentage of factors enriched and depleted at TRCs for each of the GO categories analyzed. **d** Schematic chart with the most relevant chromatin factors found enriched at TRCs. Bold text is used to highlight gene subcategories. TRC transcription-replication conflict, RF replication fork. The arrow under *RF* indicate replication fork directionality. Source data are provided as a Source data file. See also Supplementary Figs. 1 and 2.

CTCFL testis-specific insulator binding factor or the PRMT5 arginine N-methyltransferase 5, among others.

Next, we investigated whether the role of the identified factors was more relevant in head-on versus co-directional TRCs. A detailed comparison of factor abundance revealed that 196 showed higher enrichment at head-on collisions, while only 5 were more abundant at co-directional TRCs (Supplementary Fig. 2a, b and Supplementary Tables 3 and 4). Factors enriched at head-on TRCs covered all GO categories previously identified (Supplementary Fig. 2b and

Supplementary Table 3). In contrast, TRC-depleted proteins were found more abundant in co-directional TRCs (Supplementary Fig. 2c, d and Supplementary Table 4). These included the replicative helicases MCM2/3/5/7 and factors involved in chromatin organization like H3K27me3, PRMT5, CTCFL or SUZ12.

Altogether, our data supports the view that TRCs, besides triggering a strong DDR associated with changes in the presence of replication and DSB repair factors, are linked to chromatin modification and remodeling, especially at head-on TRCs. Replication might be

hindered leading to alternative mechanisms as suggested by the decrease of MCM helicases.

### Histone 3 serine 10 phosphorylation extends ~1 Mb upstream of TRCs and concentrates in head-on orientation

Our results support the idea that fine tuning of chromatin is key to prevent TRCs from resulting in DNA breaks, since chromatin structure needs to be dynamic to facilitate DNA replication and transcription and prevent DNA damage. In this context, harmful R-loops have been found associated with histone H3 Ser10 phosphorylation (H3S10pho) from yeast to mammals[42], which suggest a functional link between H3S10pho and genetic instability caused by R-loops. This conclusion is supported by the observation that yeast histone H3 mutants unable to generate H3S10pho show high levels of R-loops not accompanied by an increase in DNA damage[43]. However, how R-loop-associated H3S10pho determines R-loop harmfulness is unclear. Given the strong abundance of chromatin-modulating factors at TRCs, particularly head-on TRCs, we compared H3S10pho at co-directional versus head-on TRCs.

Since MYCN was recently shown to activate on chromatin Aurora-A, which phosphorylates H3S10pho in S phase and promotes R-loop suppression[44], we used the available genome-wide data of H3S10pho profiling during S phase from this study to investigate the coverage of H3S10pho at TRCs. First, we compared the H3S10pho genomic distribution in the neuroblastoma cell line IMR-5 with the DRIPc-seq data in control K562 cells (Fig. 2a, b). Consistent with published data from yeast[42,43], the genomic profile of H3S10pho during S phase correlated with that of R-loops in K562 cells. Metadata analysis showed a clear accumulation of H3S10pho at these sites: 1951 out of 2771 (70.4%) of H3S10pho-accumulating genes were also R-loop prone (Fig. 2c, d).

Furthermore, a massive and asymmetric increase of H3S10pho was detected around head-on TRCs at the Mb scale that was not observed at co-directional TRCs (Fig. 2e). H3S10pho extended from ~0.2 Mb to ~1 Mb upstream the TRC, and certain accumulation was also appreciable by ~0.5 Mb downstream. This is consistent with H3S10pho foci accumulation detected in IF experiments under conditions triggering R-loop accumulation in human cells[42]. Notably, chromatin interactions are more intense at head-on rather than co-directional

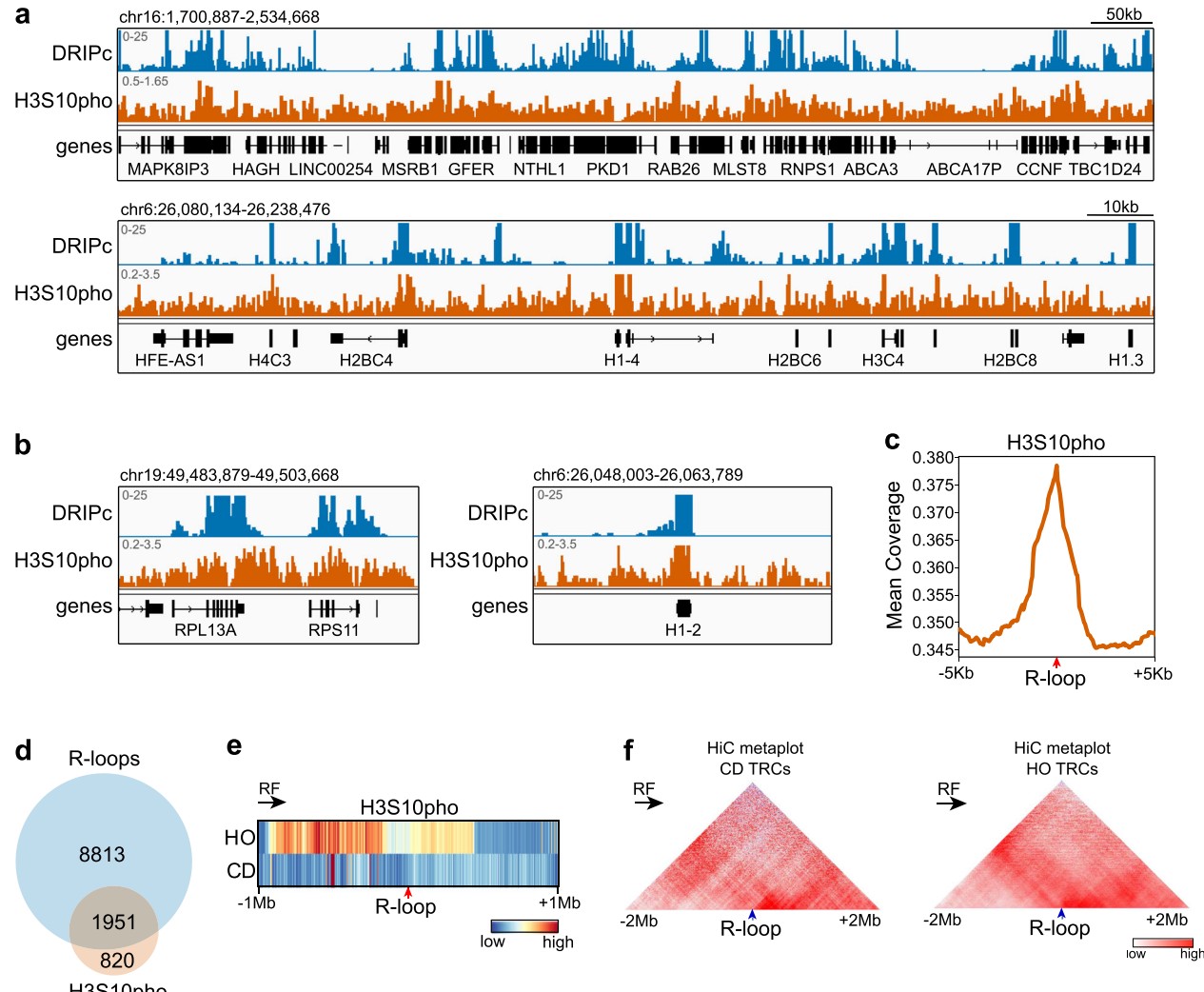

**Fig. 2 | Evaluation of H3S10pho signal over TRCs. a** Representative screenshot of a genome region showing co-localization of R-loops (blue) and H3S10pho (orange). **b** Examples of two R-loop-prone genes (*RPL13A* and *H1-2*) showing co-localization with H3S10pho. **c** Metapeak analysis. H3S10pho mean coverage around +/−5 kb of R-loop peaks. **d** Venn diagram showing co-occurrence of R-loop-prone genes (DRIPc; blue) and H3S10pho (orange) mark in control K562 cells. **e** Metanalysis at

megabase scale. H3S10pho mean coverage around +/−1 Mb of head-on and co-directional R-loop peaks. Replication fork directionality is indicated. **f** Analysis of chromatin interactions around TRCs. Replication fork directionality is indicated. TRC transcription-replication conflict, HO head-on, CD co-directional, RF replication fork. Arrows under *RF* indicate replication fork directionality.

TRCs, especially upstream of the TRCs where H3S10pho accumulates, as determined by HiC data metanalysis (Fig. 2f), consistent with the broadly reported connection between H3S10pho and chromatin condensation.

These results further confirm the connection between R-loops and H3S10pho genome-wide in human cells and reveal a major occurrence of the modification at head-on collisions.

## Depletion of TRC-enriched chromatin factors leads to R-loop-mediated DNA damage

Alteration of epigenetic factors is common in cancer cells, and their deficiencies have been associated with genome instability, TRCs being a potential source of DNA damage and genome instability. Given that the GO term with the highest enrichment scores at TRCs was ATP-dependent chromatin remodelers (Fig. 1 and Supplementary Fig. 1), we next assayed whether silencing the expression of components from the ISWI, INO80 and CHD families, whose impact on TRCs had not been analyzed previously had an impact on R-loop metabolism, as observed for the SWI/SNF family[26]. Thus, we analyzed the consequences of depleting SMARCA5 and INO80, the core subunits of chromatin remodelers ISWI and INO80, and MTA2, a subunit of the Nucleosome Remodeling and Deacetylase (NuRD) complex that belongs to the CHD remodeling family, on genome integrity.

We depleted SMARCA5, INO80 and MTA2 in HeLa cells via small interference RNA (siRNA) that caused >80% depletion efficiency as determined by Western Blot (WB) (Supplementary Fig. 3a, b). Then, we analyzed DNA break accumulation by immunofluorescence (IF) using the γH2AX antibody as a marker. In most cases, gene silencing resulted in an accumulation of γH2AX foci consistent with an increase in DNA breaks (Fig. 3a and Supplementary Fig. 3c, d), as reported in siSMARCA4 cells[26]. The percentage (%) of cells with more than 5 γH2AX foci raised significantly by ~1.6× in siSMARCA5, siINO80 and siMTA2 cells compared to siRNA control cells (siC). Next, to unveil whether the increase on DNA damage formation was due to abnormal R-loop metabolism, we overexpressed RNase H1 (RNH1), which degrades the RNA moiety of DNA-RNA hybrids, and measured the impact on DNA break occurrence. The results showed a significant decrease in the % of cells containing more than 5 γH2AX foci, close to the levels of siC cells in all the three conditions showing increased DNA damage upon gene depletion, supporting an R-loop-dependent formation of DNA breaks (Fig. 3a and Supplementary Fig. 3c, d).

Next, we assayed whether the levels of R-loops were increased, as determined by IF using the anti-DNA-RNA S9.6 antibody in siINO80 and siSMARCA5 cells that showed the highest γH2AX foci accumulation. Consistently, the mean nuclear S9.6 IF signal was significantly higher in siSMARCA5 (26.3) and siINO80 (34.5) cells compared to siC control cells (19.6), with a nucleolar increase of S9.6 signal also detectable, especially in siINO80 cells (Fig. 3b and Supplementary Fig. 3e, f). Importantly, the S9.6 signal was significantly reduced by overexpressing RNH1, (Fig. 3b and Supplementary Fig. 3e, f), confirming the specificity of detection of DNA−RNA hybrids. Concomitantly, R-loops were also assayed by DNA−RNA immunoprecipitation (DRIP) followed by quantitative PCR (qPCR). In accordance with the results coming from IF experiments, the precipitated material measured as % of input was significantly increased in RNAPII-transcribed genes (*RPL13A*, *TAF9B* and *FOXP4* in siSMARCA5 cells and *RPL13A* and *FOXP4* in siINO80 cells) when compared to siC cells (Fig. 3c). Moreover, the quantity of immunoprecipitated material was also significantly higher for rDNA genes (*28 S* in siSMARCA5 and *5′ region* and *28 S* in siINO80), coincident with nucleolar IF results (Supplementary Fig. 3f). In all cases, the DNA-RNA hybrid signal was reduced dramatically by treating the samples with ribonuclease H (RNH) prior to immunoprecipitation, confirming again that results referred specifically to DNA-RNA hybrids (Fig. 3c).

Altogether, these results show that in addition to SWI/SNF, chromatin remodeler factors such as INO80, SMARCA5 and MTA2 control harmful R-loop accumulation in cells, implying a global role of the chromatin remodeling network in R-loop homeostasis and associated genome instability.

## Chromatin alteration causes R-loop-associated DNA replication impairment and S-phase DNA damage

R-loops toxicity is in most cases linked to their ability to block DNA replication during S phase[4]. Therefore, we studied whether unscheduled R-loop formation in siSMARCA5 and siINO80 cells resulted in TRCs, stalled forks and S phase-associated DNA damage.

We first investigated TRC occurrence in SMARCA5 and INO80 depleted cells by Proximity Ligation Assay (PLA) using antibodies against PCNA and the elongating form of RNA polymerase II (RNAPII) phosphorylated at S2 (RNAPII-S2P), as previously described[26]. RNAPII-S2P + PCNA PLA foci median values increased significantly by 1.75-fold in both siSMARCA5 and siINO80 cells (Fig. 4a). Consistently, increased fork stalling and DNA damage during S phase were also detected in both conditions, as determined by IF of FANCD2 foci using an anti-FANCD2 antibody, and the analysis of DNA break accumulation along the cell cycle by IF of γH2AX foci, using DAPI staining to infer cell cycle phase by the DNA content, as previously established[26]. Indeed, a significant 2-fold increase in the % of cells with more than 5 FANCD2 foci and median values of γH2AX foci per cell in S phase were observed in SMARCA5 and INO80-depleted cells (Fig. 4b and Supplementary Fig. 4a). Notably, all the observed phenotypes were RNH1-sensitive; the increase in RNAPII-S2P + PCNA PLAs, FANCD2 and S-phase γH2AX foci were significantly suppressed by RNH1 overexpression. These results support that unscheduled R-loop formation boosts TRCs leading to increased fork stalling and DNA breaks (Fig. 4a, b and Supplementary Fig. 4a).

To further investigate the function of these chromatin factors at R-loops and TRCs, we analyzed the genome-wide distribution of SMARCA5 and YY1 subunit of INO80 and observed a clear overlap with R-loops and FANCD2 enriched regions, previously determined by DRIPc-seq and ChIP-seq in K562 cells (Fig. 4c, d). Indeed, 3339, 1950 and 816 R-loop peaks colocalized with SMARCA4, SMARCA5 and YY1, respectively, being FANCD2 clearly enriched over these sites, as shown by metaplot analysis of FANCD2 ChIP-seq (Fig. 4e, f). At the gene level, 68.0% target genes of SMARCA5 and 64.0% target genes of YY1 are R-loop-positive genes. Of them, 39.4% and 38.3% are also enriched in FANCD2 (Fig. 4g, h). Remarkably, 81.3% FANCD2-target genes are enriched in SMARCA5 and INO80 proteins, as deduced from ChIP-seq (Supplementary Fig. 4b).

Strikingly, despite their general association with R-loops and TRCs, chromatin remodelers mostly associate with specific and mutually exclusive R-loop subsets. Indeed, only 247 R-loops showed enrichment in all three remodeling activities, while 1927, 660 and 299 R-loops colocalized only with SMARCA4, SMARCA5 and YY1, respectively (Fig. 4e). Similarly, metapeak analysis unveiled clearly different profiles of each chromatin remodeler over the R-loop regions. While SMARCA4 shows a clear asymmetric distribution, consistent with previous data, SMARCA5 and YY1 accumulate with a sharp profile at the R-loop sites (Supplementary Fig. 4c). In addition, YY1 is enriched asymmetrically with a gradual increase along the transcription unit until reaching the R-loop prone site. Peak annotations also revealed that YY1-enriched R-loops are at promoter and 5′ UTR regions, while R-loops enriched in SMARCA4 and SMARCA5 are enhanced at intronic and intergenic areas (Supplementary Fig. 4d).

The R-loop-prone genes enriched in chromatin remodelers do not overlap and display specific features. Among these genes, 248 are bound by all three remodelers, whereas 1096, 371 and 215 only present peaks of SMARCA4, SMARCA5 or YY1, respectively (Supplementary Fig. 4e). Such genes present high expression values when compared to the whole genome, consistent with a higher chance of R-loop

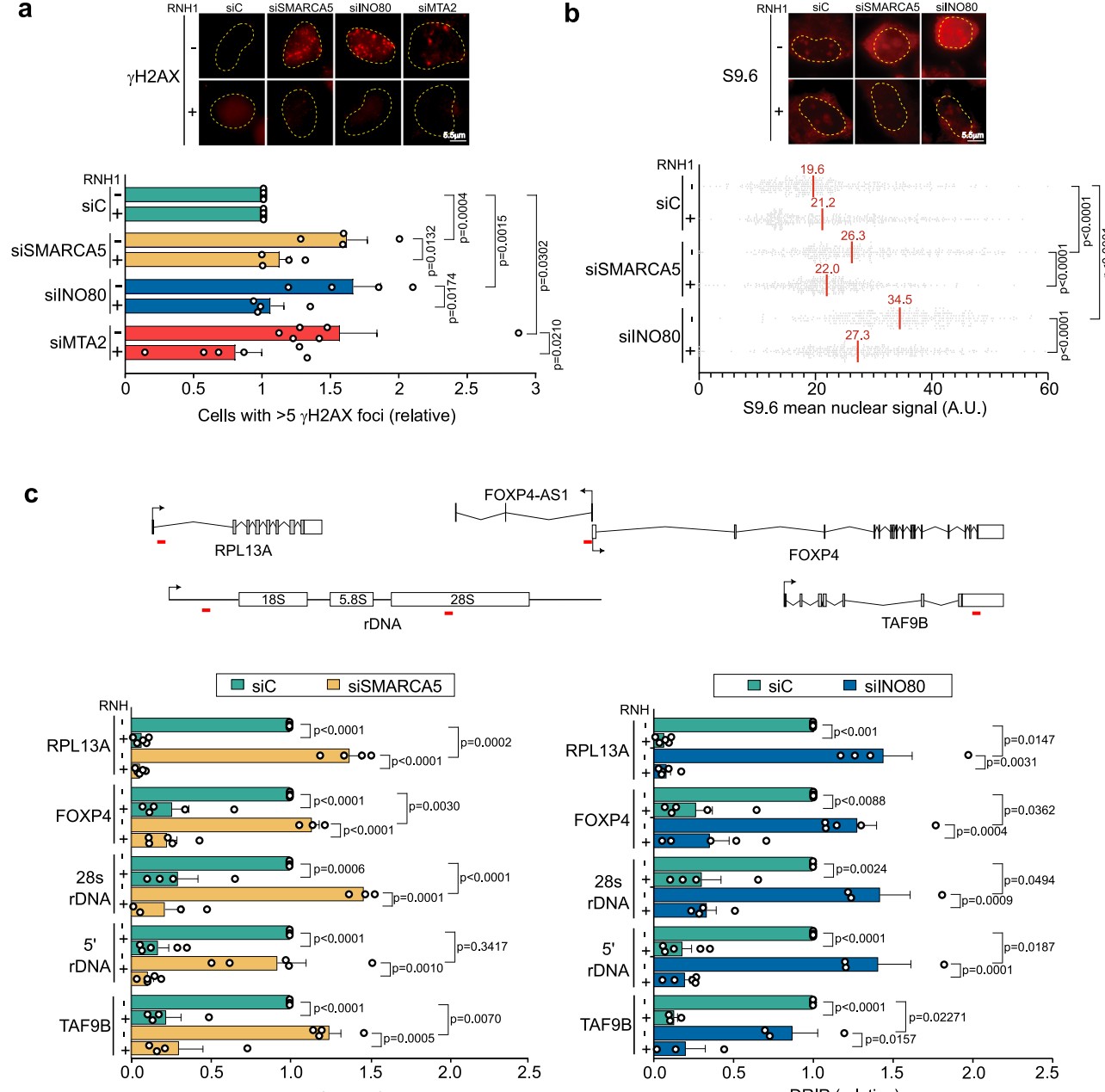

**Fig. 3 | Assessment of DNA damage in siSMARCA5, siINO80, and siMTA2 cells.**
**a** Percentage of cells with >5 γH2AX foci in control (siC) and SMARCA5, INO80, MTA2-depleted cells that overexpress (+) or not (−) RNH1. Data expressed as relative to siC. Mean + SEM are plotted ($n = 4$ (siSMARCA5 and siINO80) and n = 6 (siC and siMTA2) independent experiments). (Unpaired Student's $t$ test, one-tailed). **b** Quantification of nuclear S9.6 mean signal intensity in siSMARCA5 and siINO80 cells treated as in (**a**). Data presented as scatter plot ($n > 100$ cells examined over 3 independent experiments). Median values are indicated. (Mann–Whitney $U$-test, two-tailed). **c** DRIP-qPCR analysis of RNAPII (*RPL13A*, *FOXP4*, *TAF9B*) and RNAPI-transcribed genes (5' and *28S rDNA*) of siC, siSMARCA5 (left) and siINO80 (right) cells. Signal values normalized with respect to the siC control and plotted as mean ± SEM ($n = 5$ (*RPL13A*, *FOXP4* and *5' rDNA*) and $n = 4$ (*TAF9B* and *28S rDNA*) independent experiments). (Unpaired Student's $t$ test, one-tailed). Representative images, with nuclear perimeter highlighted (yellow dashed line), are shown. Gene regions amplified by qPCR in DRIP-qPCR experiments are indicated with a red line on drawings of the genes tested. Scale bars and $p$ values are indicated. Source data are provided as a Source data file. See also Supplementary Fig. 3.

formation. Nevertheless, the properties of each gene subset are distinct. Thus, R-loop-prone genes enriched in SMARCA4 are significantly longer, SMARCA5-enriched R-loop genes display higher GC content, and YY1-enriched genes show higher expression levels (Supplementary Fig. 4f–h).

These results support the view that chromatin remodelers prevent TRCs from resulting in DNA damage during S phase, as reported for the BAF subcomplex of SWI/SNF[26,30], under clearly distinct genome contexts.

## R-loop-prone sites correlate with somatic mutation hotspots in cancer genomes

Cancer is a multifactorial disorder where genomic alterations are recurrent, chromatin malfunctioning having a direct impact on the evolution and prognosis of the disease as evidenced by the high number of alterations found at chromatin-modulating genes in malignant cells[45]. The variety of proteins enriched at genomic regions with high potential to accumulate TRCs, known to be stimulated by R-loops, opens the possibility that such regions are potential hotspots of

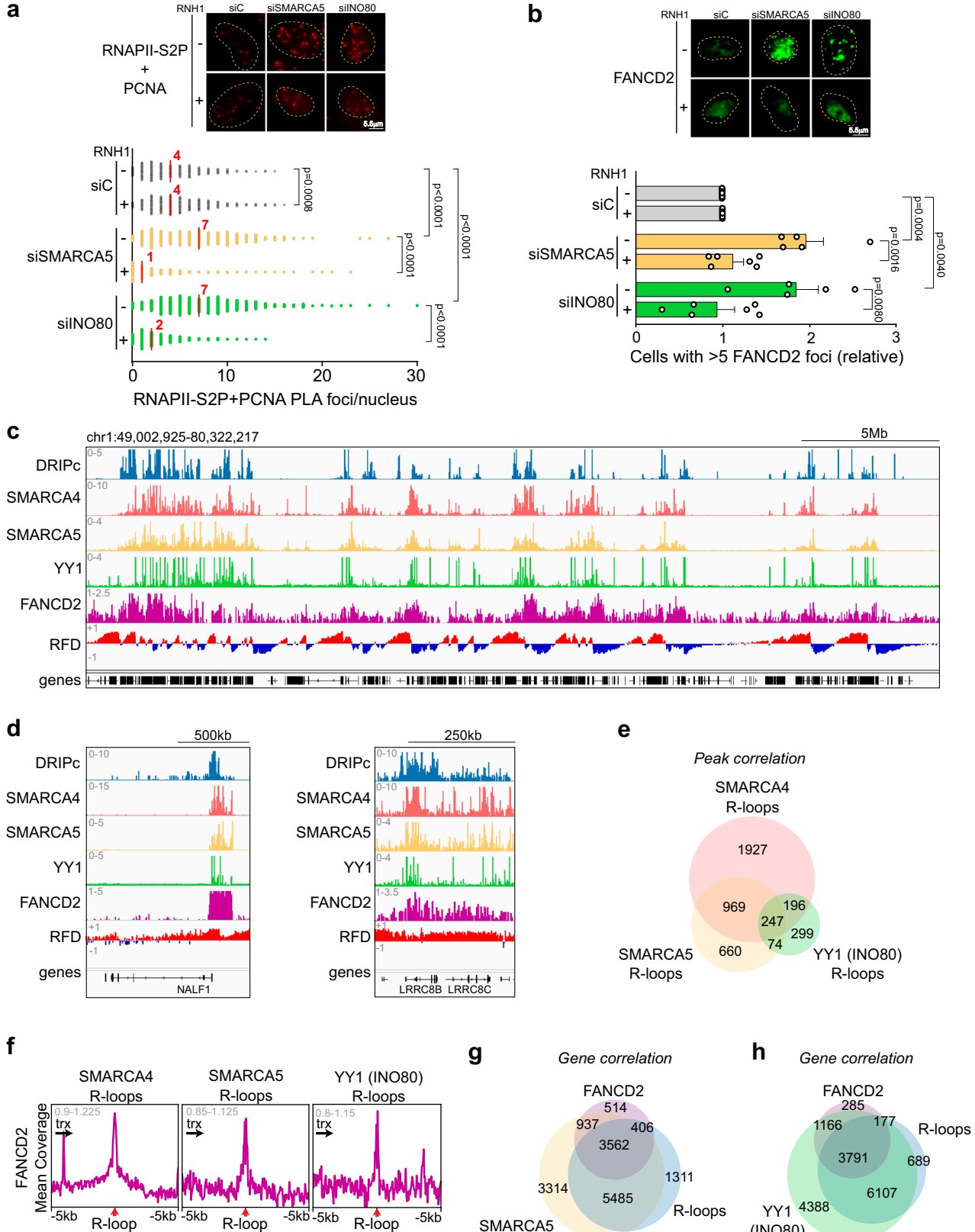

mutations which could be specifically manifested in cancer cells, consistent with the specific mutation signatures and their potential instigators identified in transformed cells[46]. Therefore, an analysis of the mutation patterns of R-loop-prone sites might provide insight into the possible mechanisms of R-loop-induced genetic instability, where chromatin dynamics might play relevant roles.

We crossed the mutation data from the COSMIC[24], a database holding extensive curated somatic mutation information related to human cancers, with genome-wide R-loops obtained by DRIPc-seq[26]. We first gathered coding and non-coding mutation data from the COSMIC database and divided mutations into SNVs, deletions and insertions and analyzed their abundance at R-loop-hot regions versus

**Fig. 4 | Analysis of DNA damage along the cell cycle in siINO80 and siSMARCA5 cells. a** Quantification of RNAPII-S2P + PCNA PLA (TRCs) foci in control (siC), siSMARCA5 and siINO80 cells. Data presented as scatter plot ($n$ > 50 cells examined over 3 independent experiments). Median values are indicated. (Mann–Whitney U test, two-tailed). **b** Percentage of cells with more than 5 FANCD2 foci in siC, siSMARCA5 and siINO80 cells that overexpress (+) or not (−) RNH1. Data expressed as relative to siC. Mean + SEM are plotted ($n$ = 5 (RNH1−) and $n$ = 6 (RNH1+) independent experiments). (Unpaired Student's $t$ test, one-tailed). **c** Representative screenshot of a genome region showing co-localization of R-loops (DRIPc; light blue), SMARCA5 (yellow), YY1 (INO80; green), SMARCA4 (red) and FANCD2 (purple) at sites with high R-loop abundance. Replication fork directionality (RFD) is also shown. **d** Zoomed-in examples of R-loops (DRIPc; light blue), SMARCA5 (yellow), YY1 (INO80; green), SMARCA4 (red) and FANCD2 (purple) colocalization. **e** Venn diagram showing correlation between SMARCA4 (red), SMARCA5 (yellow) and YY1 (INO80; green) peaks colocalizing with R-loops. **f** Metapeak analysis of FANCD2 at R-loops colocalizing with SMARCA4, SMARCA5 or YY1. Transcription direction is indicated. **g** Venn diagram showing R-loop (DRIPc; light blue), FANCD2 (purple) and SMARCA5 (yellow) genome-wide co-occurrence in control K562 cells. **h** Venn diagram showing R-loop (DRIPc; light blue), FANCD2 (purple) and YY1 (INO80; green) genome-wide co-occurrence in control K562 cells. Representative images, with nuclear perimeter highlighted (yellow dashed line), are shown. Scale bars and $p$ values are indicated. RFD replication fork directionality, trx transcription. Arrows under *trx* indicate transcription direction. Source data are provided as a Source data file. See also Supplementary Fig. 4.

other regions (Fig. 5a–c). Metadata analysis resulted in noticeable SNV, deletion and insertion metapeaks at R-loop sites (Fig. 5d), suggesting that R-loops are an important source of mutagenesis in cancer cells. Indeed, the mutation burden was significantly higher over R-loop areas compared to the mutation load of the same genomic ranges being randomized along the same genes (Supplementary Fig. 5a). Consistently, mutations were clearly detectable at R-loop-prone genes, such as those routinely tested by DRIP-qPCR (*RPL13A*, *EGR1*) (Fig. 5b, c). Furthermore, the mutation profiles showed clear enrichments over 5' and 3' end of R-loop-prone genes (Fig. 5e), as also reported for R-loops.

Given the tight connection observed between R-loops and mutagenesis, we also measured the impact of R-loop formation on mutagenesis genome-wide to gain further insight into the mutagenic consequences of these structures in cancer. We first determined mutation hotspots genome-wide by using MACS2 algorithm[47], a computational method commonly used to detect genome regions with coverage enrichments (Supplementary Fig. 5b). Then, we crossed mutation-enriched sites with genes and classified these according to expression and R-loop accumulation in K562 cells. Strikingly, 53.7% of mutation hotspots colocalize with R-loop-prone genes, while only 25.7% and 12.6% coincide with R-loop-reluctant and silenced genes, respectively (Fig. 5f). Further statistical analysis revealed a significant association between R-loop-prone genes and mutation hotspots (Supplemental Fig. 5c).

We next compared the impact of mutagenesis in head-on and co-directional R-loop-associated TRCs. Either SNVs, deletions or insertions were found significantly higher at head-on TRCs, consistent with these being the most important compromising genome integrity[26,28,41,48–50]. Indeed, the number of R-loop-prone sites with more than 1 SNV/kb significantly rises from 76.9% in co-directional to 81.1% in head-on TRCs, while R-loops with >0.1 indel/kb increases from 20.9–23,6% in co-directional to 27.4–28.2% in head-on TRCs (Fig. 5g). Consistently, SNVs and indels were much more abundant at sites accumulating unscheduled R-loops after depletion of SMARCA4 than of the UAP56/DDX39B DNA-RNA helicase (Supplementary Fig. 5d), in accordance with SMARCA4 but not UAP56 playing a key role helping resolve TRCs[26]. These results are consistent with a major genotoxicity of head-on TRCs, as widely reported.

Finally, we analyzed the impact of chromatin remodeling deficiencies on R-loop mutation burden. We selected data from tumors with at least 1000 mutations (9,046 tumors with 23,013,925 SNVs and 1,636,758 indels) and sorted COSMIC tumor samples based on the status of chromatin remodeler genes; then we compared the mutation load of samples carrying disruptive chromatin remodeler mutations to those with proficient chromatin remodeler genes. Strikingly, R-loop mutation burden was significantly higher in tumors holding *SMARCA4*, *SMARCA5* and *INO80* gene deficiencies (Fig. 5h), consistent with a direct role of these activities preventing R-loop mutagenesis in cancer.

Altogether, these results are consistent with a major role of chromatin remodelers helping prevent mutation burden at R-loop-prone regions in cancer cells.

## Chromatin patterns associated with R-loop-prone mutational signatures

Mutational processes might be driven by different etiologies during cancer development, each one causing specific and identifiable mutational patterns commonly referred as mutational signatures. Therefore, we next deepened into the potential causes of R-loop mutagenesis boosts observed in cancer cells and their association to the epigenetic context.

First, we analyzed the single-nucleotide variant (SNV) and insertion–deletion (indel) mutational signatures associated to these sites. We used *SigProfiler*[51] to identify the mutational patterns and its similarity to previously described signatures in the mutation dataset. The result revealed 17 SNV and 13 indel mutational signatures (Supplementary Figs. 6 and 7 and Supplementary Tables 5 and 6) matching a wide range of mutation etiologies. Then, we computed the number of mutations corresponding to each signature present at R-loop sites and performed metaplot analysis of them around +/−50 kb from the R-loop peak centers. Notably, mutation signatures peaked at R-loop sites, being those associated with dysfunction of DNA repair processes the most abundant. Indeed, mutations resulting from defects in homologous recombination (HR; *BRCA1* and *BRCA2* mutations), transcription-coupled Nucleotide Excision Repair (NER-TCR), DNA mismatch repair (MMR), translesion synthesis (TLS; polymerase η), base excision repair (BER), topoisomerases (topoisomerase 1 and 2A), non-homologous end joining (NHEJ), replication slippage and AID/APOBED-mediated cytidine deamination scored high at these sites (Fig. 6a and Supplementary Fig. 8a, b). Consistently, deregulation of several factors involved in these processes, including BRCA1, BRCA2 and TOP1 induces R-loop-dependent genome instability[9,19,20,52]. Remarkably, mutations associated with HR-defects and replication slippage were among the most abundant (Fig. 6a and Supplementary Fig. 8a, b), suggesting that fork stalling at TRCs may be a major source of mutagenesis at these regions.

Importantly, mutational signature comparisons unveiled a major impact of HR dysfunction, NER-TCR and the AID/APOBEC family of cytidine deaminases on head-on TRCs in cancer cells (Fig. 6a and Supplementary Figs. 9 and 10). Similarly, signatures analogous to those induced by reactive oxygen species (ROS) or aristolochic acid exposure, as well as one with unknown etiology characterized by increased >5 bp insertion mutations were also higher at head-on TRCs. In addition, despite not reaching significant differences, a higher abundance of mutational footprints related to translesion synthesis (TLS) polymerase η or topoisomerase was also appreciable at head-on TRCs.

We next investigated the association of chromatin factors enriched at TRCs in our screening with the different mutational patterns peaking at R-loop-prone regions. To this aim, we first measured the frequency of mutational signatures at R-loop-prone sequences enriched for each of the most relevant TRC-enhanced epigenetic factors identified (Fig. 1d) and compared it to the frequency at all R-loop-prone genomic regions. The results unveiled differential associations that were grouped into 7 clusters (Supplementary Fig. 11). Consistent with previous data[26], SWI/SNF complex factors SMARCA4 and ARID1B

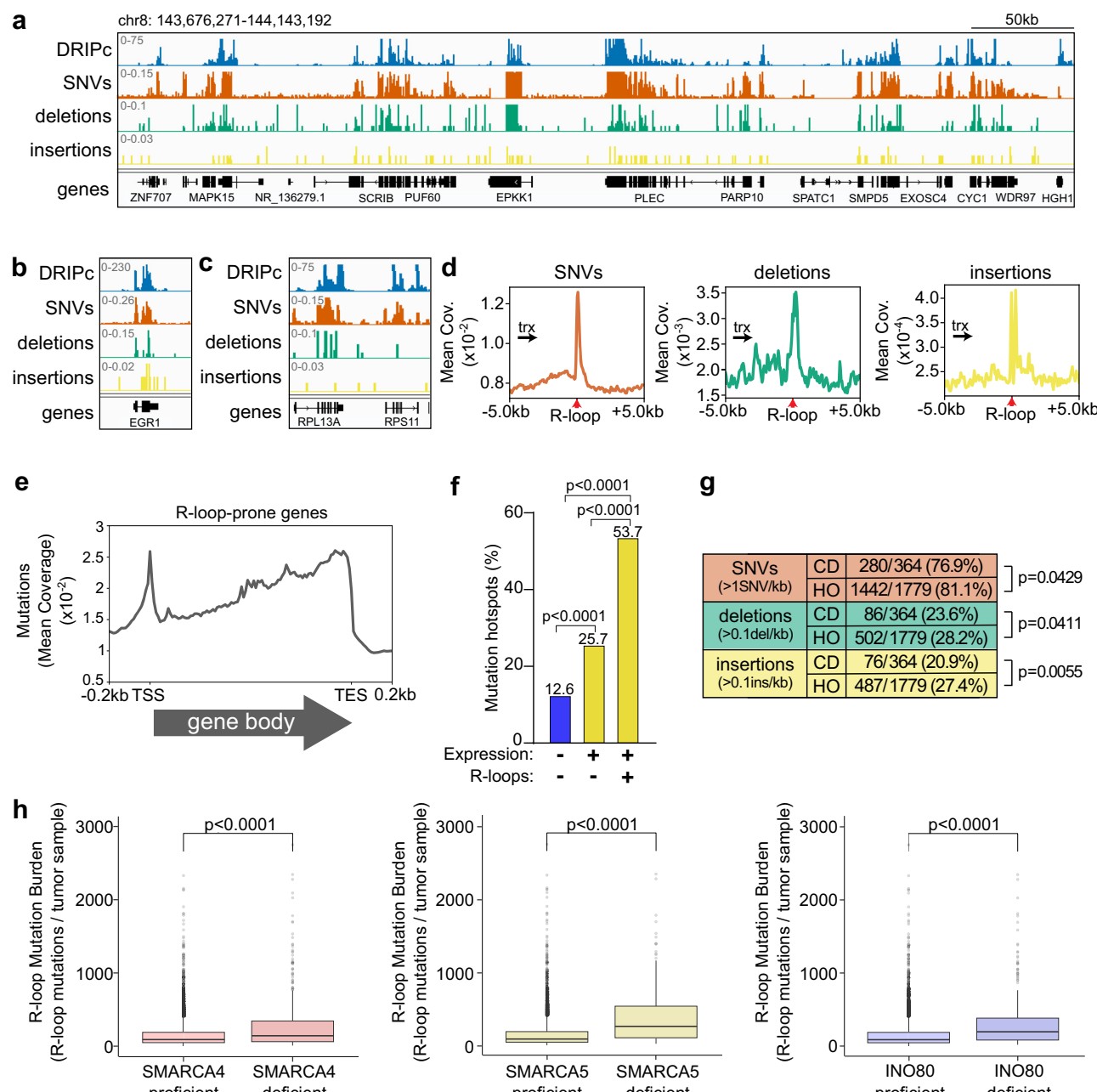

**Fig. 5 | Analysis of the mutational landscape at TRCs in cancer. a** Representative screenshot of a genome region showing co-localization of R-loops (DRIPc; blue) with SNVs (orange), deletions (green) and insertions (yellow). **b** Example of an R-loop-prone gene (*EGR1*) showing co-localization with SNVs (orange), deletions (green) and insertions (yellow). **c** Example of an R-loop-prone gene (*RPL13A*) showing co-localization with SNVs (orange), deletions (green) and insertions (yellow). **d** Mutation metanalysis. Mean coverage of SNVs (orange), deletions (green) and insertions (yellow) around +/− 5 kb of R-loop peaks. Transcription direction is indicated. **e** Metagene analysis of mutations over R-loop-prone genes. The arrow below the graph indicates transcription direction. **f** Percentage of mutation hotspots in cancer colocalizing with R-loop-prone genes, expressed genes reluctant to R-loop formation and silenced genes. (Chi-square with Yates' correction test, two-tailed). **g** Quantification of the number of sites with >1SNV/kb, >0.1 deletions/kb

and >0.1 insertions/kb at head-on and co-directional TRCs. (Fisher's exact test, one-tailed). **h** Comparison of R-loop mutation burden between tumor samples proficient and deficient in SMARCA4 (left panel), SMARCA5 (middle panel) and INO80 (right panel). Data presented as box plot. The lower and upper hinges of the box plots correspond to the first and third quartiles (the 25th and 75th percentiles), while the upper and lower whiskers extend from the hinge to the largest and smaller value no further than 1.5× inter-quartile range, respectively. Data beyond the end of the whiskers are considered outliers and plotted individually. *n* = 9046 biologically independent tumors sorted according to gene status. (Wilcoxon test, two-tailed). Scales and *p* values are indicated. SNV single-nucleotide variant, HO head-on, CD co-directional, trx transcription. Arrows under *trx* indicate transcription direction. Source data are provided as a Source data file. See also Supplementary Fig. 5.

were clustered together and enriched at sequences with high SNV signatures driven by APOBEC mutagenesis and HR deficiencies and sequences with indel patterns observed at sites of head-on TRCs. In contrast, SMARCA5 and MTA1/2/3 strongly associated with APOBEC mutagenesis, MMR deficiencies and topoisomerase 1/2A-dependent

mutagenesis, as also observed for H4K20me1, a histone modification tightly associated with DNA repair and replication[53]. Other factors including YY1, an INO80 subunit, or SIN3B HDACs correlated with APOBEC-induced mutagenesis, MMR deficiencies, DNA replication slippage and different patterns of unknown etiology. Strikingly,

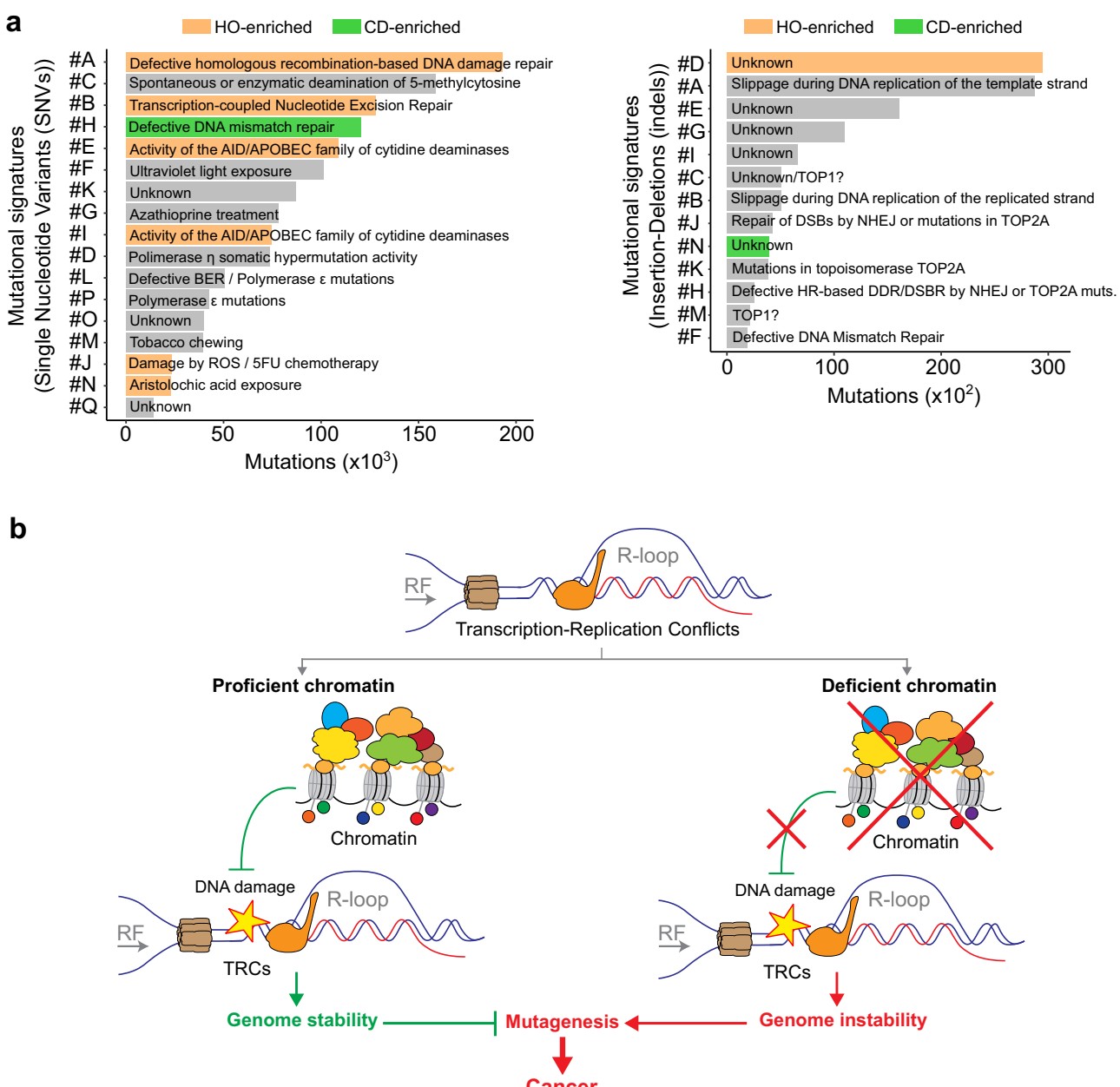

**Fig. 6 | Study of the R-loop-associated mutational signatures. a** Quantification of Single Nucleotide Variants (SNVs; left) and insertion–deletions (indels; right) associated to R-loops corresponding to each of the mutational signatures. Head-on enriched signatures are highlighted in orange, and co-directional enriched in green. **b** Model. Schematic representation highlighting that a proficient chromatin is required for a proper response to transcription-replication conflicts. Under chromatin-deficient scenarios, transcription-replication conflicts pose a source of DNA damage and hot-spot for mutagenesis that might predispose cells to transformation or help them surpass biological barriers during cancer development. SNV single-nucleotide variant, indel insertion–deletion, HO head-on, CD co-directional, RF replication fork. Arrows under *RF* indicate replication fork directionality. See also Supplementary Figs. 6–11.

H3S10pho formed a clearly differentiated cluster with tight associations with TC-NER, defective HR, APOBEC-mutagenesis, polymerase η and topoisomerase 2A, consistent with a major role at TRCs. Furthermore, FOXA1, a transcription factor involved in chromatin remodeling and associated with DNA repair complexes, that is required for genomic targeting of DNA polymerase β (POLB) in human cells[54,55], also showed specific association patterns. FOXA1 was enriched at regions with signatures associated with head-on TRCs (defective HR, TC-NER, ROS) as well as with DNA replication slippage and topoisomerase 1 and 2A-driven mutagenesis, unveiling a crucial connection between FOXA1 and TRC-mutagenesis.

Overall, these results are consistent with a differential association of chromatin factors with mutagenic DNA processes, highlighting the importance of epigenetics in R-loop-associated mutagenesis and genome instability.

## Discussion

TRCs may be an important source of genome instability, especially in cancer cells[1,4]. Deciphering the factors that orchestrate TRCs regulation and their mechanisms might help understanding how these toxic events occur and identify new therapeutic approaches in the clinic. Our approach unveils a major contribution of chromatin factors on

TRCs resolution, especially in head-on collisions, which are more difficult to solve than co-directional TRCs and therefore prone to replication fork breakage and DNA damage[4], as supported by the presence of fork-processing proteins and DDR factors at these sites (Fig. 1 and Supplementary Figs. 1 and 2). It is worth noticing that DNA-RNA hybrids are also induced by DSBs[56], which may open the possibility that part of the accumulation of hybrids could be caused by DSBs. However, this is not the case here since RNH1 overexpression suppresses DNA breaks, as expected if these are a consequence of the hybrids.

At TRCs, fine-tuning of chromatin is critical, as deduced by the large number of chromatin modifiers, remodelers and histone PTMs observed to concentrate at these sites (Fig. 1). These factors might directly impact TRC resolution through different mechanisms, ensuring genome stability and cell viability. Consistent with this, specific histone modifications such as histone acetylation or methylation has been previously associated with transcription-dependent DNA damage[32,57,58]. Similarly, previous evidence supports that the corresponding histone modifiers as is the case of BRD4 that binds acetylated histone residues are enriched at TRCs. In this context, it is worth noticing the high abundance of H3K79me2 at TRC sites, pointing to a crucial role of this particular histone modification during this process. Indeed, this histone PTM is closely linked to the DDR, as it is required for Rad9 recruitment in *S. cerevisiae*[59–61] and for 53BP1 recruitment in mammalian cells, being particularly abundant at transcribed genes[59,62,63]. Moreover, the writer of H3K79me2, DOT1L, is frequently altered in some cancer types[64], suggesting a role of such epigenetic mark on genome integrity. In contrast, H3K36me3, another histone PTM enriched at transcribed genes[65], is not found enriched at head-on TRCs, which suggests that H3K79me2 at TRCs might not respond to a cause-effect relationship. Interestingly, H3K27me3 seems reluctant to TRCs, which may be explained by the fact that is a repressive mark[65] and TRCs rely on active transcription. Nevertheless, it is worth noticing that H3K27 tri-methylation by EZH2 is induced at stalled forks to allow MUS81 binding and promote fork degradation and replication restart in *BRCA2*-deficient cells[66]. It is likely that the maintenance of low levels of MUS81 at stalled forks by promoting low levels of H3K27me3 might favor fork stability at TRCs.

H3S10pho might also play a relevant role at TRCs. It accumulates at R-loop sites, H3S10pho foci being clearly visible upon induction of unscheduled R-loops[42]. Parallelly, a mutation of the residue that prevents such a phosphorylation results in an increase of R-loop not accompanied by DNA breaks[43], suggesting that H3S10pho-dependent chromatin modification is a determinant of R-loop-mediated DNA damage. Interestingly, our analysis reveals a major accumulation of H3S10pho at head-on TRCs (Fig. 2). This result confirms an association between TRCs and H3S10pho, and opens the possibility that the TRC itself promotes H3S10pho. H3S10pho enrichment at head-on TRCs could be explained if the mark remains longer at these sites. TRC-associated H3S10pho could eventually compromise fork integrity or interfere with the DDR leading to DNA breaks. Further investigation is required to fully understand this process and the biological function of H3S10 phosphorylation associated with harmful R-loops.

Chromatin remodeling may play crucial roles given that SWI/SNF, ISWI, INO80 and CHD remodeling families' subunits are found significantly enriched at head-on TRCs, as also reported for the SWI/SNF complex[26]. Our results add a new key regulatory role of chromatin remodelers at TRCs. We show that SWI/SNF, ISWI and INO80 ATP-dependent chromatin remodeling families exert specific and differential functions preventing TRC-mediated DNA damage and mutagenesis. Indeed, TRCs and DNA damage increases significantly when SMARCA5 or INO80 are knocked-down in HeLa cells, being these increments of DNA breaks sensitive to RNH1 overexpression and thus R-loop-dependent. Furthermore, major increases of DNA damage occur during S/G2 phases and RNH1-sensitive increases of FANCD2 foci are appreciable in SMARCA5 and INO80-depleted cells, consistent with

a direct impact of such remodelers on DNA replication stress and TRCs. Indeed, FANCD2 correlates genome-wide with R-loops, SMARCA5 and INO80 further supporting this view. Similar phenotypes were previously described also for SMARCA4 and the SWI/SNF complex[26].

Our genome-wide analysis of mutations in cancer further supports that epigenetic regulation might contribute to the DDR triggered by TRCs. The fact that the mutational signatures associated with deficiencies in DNA repair, in particular HR and NER-TCR, and AID/APOBEC activity accumulate at head-on TRCs in cancer cells (Fig. 5), suggests a relevant role for these processes and factors in maintaining the integrity of the sites. Consistently, we observed a high RAD51 accumulation at head-on TRCs. In this context, transcription-induced hyper-recombination phenotypes due to transcription dysfunction have been broadly reported, and increased recombination at head-on collisions has also been proved in yeast using artificial systems[41]. Furthermore, targeting of the displaced ssDNA strand of R-loops or excessive DNA supercoiling may also further enhance these hazardous scenarios[67]. In this sense, C>U conversions by AID/APOBEC proteins may be targeted by the BER machinery leading to DNA nicks, that can be further processed into DSBs by the MMR pathway[67]. On the other hand, ssDNA gaps generated during the Fanconi Anemia response, which strongly impacts TRC resolution, may also be repaired by TLS, as FANCD2 recruits DNA polymerase η, a low fidelity polymerase enriched at actively transcribed genes[68,69]. Indeed, mutational patterns associated with DNA polymerase η activity are also slightly enhanced at head-on TRCs. Consistent with this, canonical replicative helicases are decreased at head-on TRCs, strengthening the idea of alternative mechanisms for completing DNA synthesis likely involving recombination and TLS.

In agreement with a potential role of chromatin remodeling in the prevention TRC-dependent DNA damage, *SMARCA4*-, *SMARCA5*- and *INO80*-deficient tumor samples display significantly increased R-loop mutation burden. Nevertheless, our results support the view that the chromatin remodelers tested display this function through multiple complementary mechanisms. Indeed, the profiles of chromatin remodelers around R-loops are clearly different, SMARCA4 increasing asymmetrically as transcription approaches to the R-loop site, INO80 peaking at the R-loop site with a weak asymmetry, and SMARCA5 peaking sharply at R-loop sites. In addition, R-loop sites enriched in each remodeler are distinct and display specific features. While R-loops colocalizing with SMARCA4 are mostly enriched in intronic regions, as similarly observed for SMARCA5, R-loop enriched in INO80 mostly correspond to Promoter and 5' UTR regions. Furthermore, the genes that are enriched in R-loops colocalizing with SMARCA4 are longer, while those colocalizing with SMARCA5 or INO80 display higher GC content or expression compared to the rest, respectively. These data suggest a non-redundant-specific function of each remodeling family, helping prevent TRC-mediated genome instability. Thus, SMARCA4 could prevent DNA damage and mutagenesis at common fragile sites (CFSs), where fragility is known to be driven by increased gene lengths; SMARCA5 might be relevant helping solve TRCs at high-GC content sites, which has been linked to higher rates of DSBs[70], and INO80 could increase DNA damage signaling by replacing histone variants at highly expressed sequences.

In summary, our observations support the view that chromatin function must be preserved at TRCs to prevent genome instability (Fig. 6b) as suggested by the high number of chromatin modifiers associated with these sites and mutational signatures resulting from malfunctioning of DNA repair pathways identified in cancer cells (Figs. 1–6). Interestingly, this might be linked to the high frequency of chromatin factors that are found altered in cancer, since deficiencies in chromatin-modifying enzymes may predispose to TRC-mediated mutagenesis, which eventually may alter tumor suppressors and/or oncogenes and, thus, favor cell transformation and adaptation during

cancer development. Alterations in chromatin-modifiers might also lead to resistance to anti-cancer drugs as they may provide additional mechanisms to survive drug treatment. Indeed, inhibitors of epigenetic factors are often used to fight certain cancer types. Thus, our study provides a new perspective to understand the high levels of mutation in chromatin-modulating activities in cancer that could enable future therapeutic approaches. Nevertheless, the benefits of using these compounds in oncogenic treatments remains largely unexploited.

# Methods

## Cell lines
Experiments were conducted using human female HeLa cell line obtained from American Type Culture Collection (ATCC) (ATCC Cat# CCL-2, RRID:CVCL_0030). HeLa cells were cultured in Dulbecco's modified Eagle's medium (DMEM; GIBCO) supplemented with 10% heat-inactivated fetal bovine serum (Sigma Aldrich, Merck KGaA) and 1% antibiotic–antimycotic (BioWEST) at 37 °C (5% $CO_2$).

## Protein knock-down
Knocking-down of proteins was achieved by transfecting HeLa cells with 50 nM siRNA against the tested genes using DharmaFECT 1 (Dharmacon), according to the manufacturer's instructions. ON-TARGET SMARTpool siRNAs from Dharmacon against SMARCA4 (L-010431-00), SMARCA5 (L-011478-00), INO80 (L-004176-01) and MTA2 (L-008482-00) were used to induce protein depletion. ON-TARGETplus Non-targeting Control Pool (D-001810-10) was used as control (siC). A detailed list of the siRNAs used for protein knock-down is available in Supplementary Table 7.

## Plasmid transfection
RNH1 overexpression was obtained by transfecting cells with pEGFP-M27-H1[71] plasmid at 1 μg/mL final concentration using Lipofectamine 2000 (Invitrogen)/Amaxa nucleofector kit R (Lonza), according to the manufacturer's instructions. pEGFP (Clontech) empty vectors were used as controls. For IF and WB experiments, cells were transfected either with pEGFP or pEGFP-M27-H1 after 48 h of siRNA treatment.

## Western blot
Protein extracts were subjected to Western blot following standard procedures. Membranes were incubated with anti-SMARCA5 (Abcam ab72499, 1:1000), anti-INO80 (Abcam ab105451, 1:1000), anti-MTA2 (Sigma HPA006214, 1:250), anti-GFP (Abcam ab290, 1:1000), anti-RNAseH1 (Proteintech 15606, 1:2000) and anti-vinculin (Sigma V9264, 1:5000). Uncropped blots are shown at the end of Supplementary Information file. Blot images were acquired using AMERSHAM ImageQuant 800 (GE Healthcare).

## Immunofluorescence
S9.6 immunofluorescence experiments were conducted as previously reported[26]. Briefly, cells were fixed with 100% ice-cold methanol, blocked with PBS-BSA 2% overnight at 4 °C and incubated with S9.6 (hybridoma HB-8730, 1:1000) and anti-nucleolin (Abcam ab50279, 1:2000) antibodies overnight at 4 °C. Then, coverslips were washed three times in PBS1X, and incubated with secondary antibodies (Thermo Fisher Scientific A21201, 1:1000) for 1 h at RT. Finally, cells were washed again, stained with DAPI and mounted in ProLong Gold AntiFade reagent (Invitrogen).

DNA damage and replication fork stalling were measured by γH2AX and FANCD2 immunostaining, respectively, as previously described[26]. Briefly, cells were pre-extracted and fixed with Triton X-100 0.1% + PBS 1X + formaldehyde (methanol-free) 4% for 10 min at RT, washed with PBS, permeabilized with PBS + 0.5% Triton X-100 for 5 min at RT and blocked with TBS 1X + BSA 3% + Tween-20 0.1% for

30 min at RT. Then, cells were incubated overnight at 4 °C with anti-γH2AX (Abcam ab2893, 1:1000) or anti-FANCD2 (Santa Cruz Biotechnlgy sc-20022, 1:100) in blocking solution, washed, and incubated again with the corresponding secondary antibodies (Thermo Fisher Scientific A11011 and A11029, 1:1000) for 1 h at RT. Finally, coverslips were washed again, stained with DAPI and mounted in ProLong Gold AntiFade reagent (Invitrogen).

IF images were acquired with a Leica DM6000 microscope equipped with a DFC390 camera (Leica) at ×63 magnification and LAS AX image acquisition software (Leica). FIJI (ImageJ) image processing package[72] was used for IF analysis. Nuclear mean gray value for S9.6, after subtraction of nucleolar signal, was measured for each condition. In the case of γH2AX and FANCD2, foci per cell were quantified.

## Proximity ligation assay (PLA)
PLA was conducted as previously described[26] using Duolink PLA Technology (Merck). Briefly, samples were fixed, permeabilized, and incubated with RNAPII-S2P (Bethyl A300-654A) and PCNA (Santa Cruz Biotechnology sc-56) primary antibodies as described for IF assays. PCNA and RNAPII-S2P antibodies were used at 1:500 dilution. Then, samples were incubated with PLA-specific secondary antibodies and PLA reagents according to the manufacturer's guidelines. Duolink in situ PLA probe anti-rabbit PLUS (Merck DUO92002), Duolink in situ PLA probe anti-mouse MINUS (Merck DUO92004) and Duolink-Detection Reagents Red (Merck) were used to perform the PLA reaction. Finally, nuclei were stained with DAPI, mounted in ProLong Gold AntiFade reagent (Invitrogen) and images acquired with a Leica DM6000 microscope equipped with a DFC390 camera (Leica) at ×63 magnification and LAS AX image acquisition software (Leica). PLA foci number per cell were quantified for all conditions.

## DNA-RNA immunoprecipitation (DRIP) assays
DRIP assays were conducted as previously described[26,73]. Briefly, genomic DNA was enzymatically digested with HindIII, EcoRI, XbaI, SspI and BsrGI restriction enzymes and DNA–RNA hybrids immunoprecipitated using the S9.6 antibody (hybridoma HB-8730). As control, samples were in vitro treated with RNase H (New England Biolabs M0297L). Finally, DNA-RNA hybrid immunoprecipitated fractions were measured by quantitative PCR (qPCR) using specific primers and immunoprecipitation rate expressed as Input %. Then, relative values respect siC without RNH were calculated and plotted. The primer sequences used in these experiments are provided in Supplementary Table 8.

## Immunofluorescence analysis and image processing
Images were acquired with a Leica DM6000 microscope equipped with an automated plate, a DFC390 camera (Leica) and LAS AX image acquisition software (Leica). FIJI image processing package[72] was used for image analysis and quantification. DNA cell cycle (MBF collection for ImageJ; https://imagej.nih.gov/ij/plugins/mbf/index.html) was added to the package to perform cell cycle-dependent IF analysis. Nuclear mean intensity and foci quantification were performed using FIJI homemade generated macros. Mean nuclear S9.6 signal was quantified after subtracting nucleolar signal, considered as that signal colocalizing with nucleolin. In the case of γH2AX and FANCD2, foci per cell were always quantified. In representative images, DAPI (4',6-diamidino-2-phenylindole dihydrochloride) nuclear staining is shown in blue, GFP and nucleolin in green and γH2AX and S9.6 in red. FANCD2 is shown in green.

γH2AX foci quantification along cell cycle was achieved by using wide-field images and cells assigned the cell cycle phase according to its DAPI content using DNA cell cycle plug-in (MBF collection) on FIJI[72]. In vivo validations of this plug-in were previously reported[13]. Finally, γH2AX foci were determined for cells of each cycle phase as for immunofluorescence experiments.

## Genome-wide data analysis

TRCs were determined genome-wide as previously described[26]. Briefly, replication fork directionality (RFD) was calculated from OK-seq and crossed with DRIPc-seq data aligned to hg38 in K562. DRIPc-seq peaks with average RFD > |0.75|, meaning homogeneous replication directionality over the R-loop sites among cell population, were selected for the study and further split into head-on or co-directional TRCs according to the direction of transcription and replication.

For the screening of factors enriched at TRCs, the entire collection of the ENCODE[22,23] project ChIP-seq datasets were retrieved and coverage along +/−1 Mb TRCs calculated using deepTools package[75]. A detailed full list with the accession numbers of the ENCODE ChIP-seq data used in this study is provided as Supplementary Data. Specifically, K562 signal *p* value ChIP-seq data from ≥2 biological replicates, except those cases where >1 replicate data was not available, were used. FANCD2 ChIP-seq data were retrieved from the NCBI Sequence Read Archive (SRA) repository[33] (https://www.ncbi.nlm.nih.gov/sra) under accession code PRJNA473287 and processed as previously described[26]. Then, signal intensity (mean coverage) from +/−10 kb around TRC was extracted and compared to 20 kb at −1 Mb (upstream ratio) and 20 kb at +1 Mb (downstream ratio), respectively. Factors with Rank Metric Scores (RMS) > |0.25| for both ratios were considered hits. In the case of H3S10pho ChIP-seq, sequencing reads were retrieved from the Gene Expression Omnibus repository[44] (https://www.ncbi.nlm.nih.gov/geo/) under accession number GSE144288, aligned to human reference genome hg38 using Bowtie2[76] and RPKM-normalized coverage calculated using deepTools package[75]. Head-on versus co-directional comparisons were performed by comparing the mean enrichment of each factor at both types of TRCs.

ChIP-seq peaks were also retrieved from the ENCODE database[22,23], and IDR thresholded peaks from 2 biological replicates were used for the analysis. MACS2[47] was used for H3S10pho peak calling. ChIP-seq peak annotations were assigned using ChIPseeker[77] and HOMER[78] computational tools. GC content was calculated using *geecee* tool from EMBOSS (The European Molecular Biology Open Software Suite)[74]. GSEA software[35,36] was retrieved from UCSC tools[79] and used to test gene enrichments for a selected group of GO categories according to our experimental purposes.

RNA-seq data was retrieved from previous analysis already published and treated as described[26]. Briefly, normalized read counts were assigned to genes and expression compared between conditions.

K562 HiC data for hg38 genome version were obtained from the ENCODE project database[22,23] entry ENCSR545YBD. Genome-wide mapping quality thresholded contact matrix datafiles from 4 biological replicates displaying 10 kb window sizes were retrieved from the repository and used to calculate average contact values around +/−2 Mb of head-on and co-directional R-loop sites using R (https://github.com/eusololi/combined_rloops_hic_map.git). Results were plotted using Gitools software[80], according to RFD.

## Cancer mutation analysis

We downloaded raw mutation calls from the COSMIC database[24] release v95 annotated to hg38, containing 400,690 tumors and 46,053,401 mutations. To identify mutational signatures, we considered only tumors with at least 1000 mutations. This criterium resulted in a total of 9046 tumors with 23,013,925 SNVs and 1,636,758 indels. BEDOPS[81] was used to convert vcf mutation data into bed files. Then bed files were further processed to bedgraph and bigwig using BEDTools[82] and UCSC tools[79]. Metaplots were obtained using Deeptools[75]. Mutation hotspots were determined using MACS2[47] *bdgbroadcall* option to call peaks from bedgraph outputs using a 0.25 cutoff for peak detection.

R-loop mutation burden was calculated by determining the number of mutations colocalizing with R-loop sites for each tumor sample with at least 1000 mutations (9046 tumors with 23,013,925 SNVs and 1,636,758 indels). Then, data was sorted based on gene status considering as deficient those samples carrying frameshift, missense and nonsense mutations inside the gene of study. R-loop sites randomization to analyze mutation load at R-loop sites was performed by using BEDtools *shufflebed* option. Matrix counts of the SBS96 and ID83 contexts were generated using the SigProfilerMatrixGenerator software[51]. Mutational signatures for SNVs and indels were inferred via the non-negative matrix factorization algorithm implemented in the SigProfilerExtractor software[83]. Then, we intersected mutations with all R-loops, head-on and co-directional with respect to replication leading to 1,477,722, 33,265, and 6628 mutations, respectively. For each mutation, we estimated the mutational signature that is more likely to cause the mutation by sampling a multinomial distribution of the vector of probabilities of each mutational signature per sample and mutation type (De_Novo_Mutation_Probabilities_refit.txt file from Sigprofiler). To assess fold-change mutation abundance for each region and the SNV_A signature, we used the *riskratio* function from the epitools R package[84].

The mutational signature frequencies along R-loop-prone sequences enriched for each chromatin factor were calculated by intersecting R-loop genomic regions +/−2.5 kb with the IDR thresholded ChIP-seq obtained from the ENCODE database[22,23] and comparing the signature abundances to the frequency at all R-loop-prone genomic regions. Differences in frequencies are expressed as log2 fold-change. Clustering of datasets was performed by using the dendextend R package[85].

## Statistical information

Statistical significances (*p* value) of the observed differences between conditions are indicated in figure panels. When *p* value was below 0.0001, *p* value < 0.0001 is only indicated. *p* Value cutoff for significance was set to 0.05, and every difference with *p* < 0.05 was considered as statistically significant. Student's *t* test was used for data represented as histograms. When data was presented as box plot or scatter plot, Mann–Whitney *U*-test or Wilcoxon test were performed. Fisher's exact test and Chi-square with Yates' correction test were used to evaluate the significance of frequency values between conditions. Test details are indicated in the figure legends.

Graphs were generated using R studio[86] and Prism (GraphPad Software, Inc.). Histogram bars always indicate mean values and whiskers standard error of the mean (SEM). Individual values are also plotted in histograms. The lower and upper hinges of box plots correspond to the first and third quartiles (the 25th and 75th percentiles), while the upper and lower whiskers extend from the hinge to the largest and smaller value no further than 1.5× inter-quartile range, respectively. Data beyond the end of the whiskers are considered outliers and plotted individually. In scatter plots, individual values are plotted and median values printed in red.

Numbers in Venn diagrams represent genes/peaks co-occurring between conditions, as indicated in the figure legends. Genome-wide screenshots were obtained from Integrative Genome Viewer (IGV)[87]. Scales are always indicated in the figure panels.

All experimental results presented in this manuscript were obtained from a minimum of 3 independent biological replicates. S9.6 IF experiments were performed analyzing more than 100 cells per replicate. In PLA analysis, a minimum of 50 cells per replicate were analyzed. Quantification of γH2AX foci along cell cycle phases was performed using more than 1000 cells from 3 biological replicates. In Figs. 3a, c and 4b, data is normalized to control samples (siC) and presented as the relative fold-change compared to the control.

## Resources details

For more information on reagents and resources used, please address to the Lead Contact Andrés Aguilera (aguilo@us.es).

## Lead contact

Further information and requests for resources and reagents should be directed to and will be fulfilled by the Lead Contact, Andrés Aguilera (aguilo@us.es).

## Reporting summary

Further information on research design is available in the Nature Portfolio Reporting Summary linked to this article.

## Data availability

The ChIP-seq data used in this study are available in the ENCODE project[22,23] database (https://www.encodeproject.org/), except the H3S10pho and FANCD2 ChIP-seq data[33] that are available in the Gene Expression Omnibus (GEO) (https://www.ncbi.nlm.nih.gov/geo/) repository under accession number GSE144288 and the NCBI Sequence Read Archive (SRA) (https://www.ncbi.nlm.nih.gov/sra) under accession code PRJNA473287. The accession numbers of the ENCODE ChIP-seq data used in this study are provided as Supplementary Data. The HiC data are available in the ENCODE project[22,23] database (https://www.encodeproject.org/) under accession number ENCSR545YBD. RNA-seq and DRIPc-seq data[26] are publicly available in the GEO repository under accession numbers GSE127979 and GSE154631. OK-seq public data[25] were obtained from the EMBL-EBI European Nucleotide Archive (ENA) database under accession code PRJEB25180. The cancer mutagenesis data used in this work was retrieved from the COSMIC[24] database (https://cancer.sanger.ac.uk/cosmic). Source data are provided with this paper.

## Code availability

The code generated in this study was deposited in public data repositories and is publicly available. The R script used to build the codirectional and head-on average contact matrices was deposited in the Github repository (https://github.com/eusololi/combined_rloops_hic_map.git; https://doi.org/10.5281/zenodo.8359279)[88]. FIJI macros for S9.6 mean nuclear intensity and γH2AX foci quantification[89] were deposited in the Zenodo repository (https://zenodo.org/record/8390817; https://doi.org/10.5281/zenodo.8390817).

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

## Acknowledgements

We thank F. Supek (Institute for Research in Biomedicine, Barcelona) for computational resources provided and discussions. Research was part of the project I + D + i PID2019-104270GB-I00/BMC, funded by MCIN/AEI/10.13039/501100011033/, as a main funding source, as well as grants from the European Research Council (ERC2014 AdG669898 TARLOOP) and Foundation "Vencer el Cancer". A.B.-F. was supported in part by Juan de la Cierva postdoctoral contracts (FJCI-2017-34536/IJC2020-044963-I) from the Spanish Ministry of Science and Innovation, E.H.-M. by a Marie Sklodowska-Curie Action postdoctoral contract (H2020-MSCA-IF-2020-RteRloop – 101027467) and M.E.S.-O. by a predoctoral FPI contract of the Ministry of Science and Innovation.

## Author contributions

A.B.-F. and A.A. designed the study and the experiments. A.B.-F. performed most of the experiments and the bioinformatic analysis. E.H.-M. and N.B.-F. contributed with specific experiments. I.G.-F. and M.E.S.-O. performed part of the bioinformatic analysis. A.B.-F. and A.A. wrote the manuscript. All authors read, discussed, and agreed with the final version of this manuscript.

## Competing interests

The authors declare no competing interests.
