## [Peer Review File · Nature Communications]

The chromatin network helps prevent cancer-associated mutagenesis at transcription-replication conflictsREVIEWER COMMENTS

Reviewer #1 (Remarks to the Author):

The work by Bayona-Feliu presents a really interesting analysis of mostly published datasets and ENCODE data that is focused understanding correlations of chromatin state determinants with sites of transcription-replication conflict (TRC). This builds on the authors previous work linking the BAF complex to these TRCs. The work is combined with some preliminary analysis of siRNA knockdowns of chromatin remodelers that implicate a few additional players in preventing R-loops associated with TRCs. I felt these two data sets did not go together very well. Notwithstanding the connection to chromatin, the experiments in Figure 1 and 2 feel like part of an incomplete story on INO80 and SMARCA5 regulating R-loops. The data analyses in Figure 3-6 hardly touch on specific chromatin remodelers at all, but rather analyze TRC features to show these are sites of mutagenesis and specific mutation signatures in various types of cancer. So the work is interesting but neither dataset seem complete and they are not well connected to one another. Below are some specific comments, but the major revision really needs to be either assessing roles of INO80/SMARCA5 in a story about that, or analysis of TRC features showing they are mutational hotspots in cancer (both potentially really great stories that are not given a full treatment in this manuscript).

1. The introduction states that R-loops are in most cases a threat to genome integrity. Can the authors defend this statement? The roles of R-loops as normal chromatin marks seem to outnumber the roles in genome integrity. R-loops at pause sites, terminators, and other regions seem to have normal functions. I guess they could have BOTH normal and toxic functions.

2. Related to Figure 1 how did the authors decide on these 4 chromatin targeting genes? Did they emerge from a screen or database as candidates? Or from the analyses later in the paper? There are dozens of chromatin binding and regulating factors, especially if we go into histone PTMs. The rationale for this small set of direct tests is poorly justified. Perhaps beginning with the ENCODE analysis would better justify validation of some of these chromatin remodelers.

2. Why do the authors think they see RNaseH1 sensitive DNA damage and S9.6 signal but not FANCD2 foci for INO80 depleted cells? If the story of the manuscript were to focus on INO80 or SMARCA5, additional analyses would be necessary that more directly measuring TRCs, replication dynamics or other features of genome instability that connect the data in Figure 1 to the story.

4. For figure 5 the authors note an enrichment of mutations in R-loop prone regions. As written it was not clear what the comparator data was? In other words, can the authors include clear analyses comparing genes with similar transcription frequencies and other variables that are NOT R-loop prone to make a statement about causality. Or do the features of R-loops (e.g. GC rich, highly transcribed) themselves predict mutation predisposition? It is important to determine whether the observations are truly R-loop specific.

5. I was not really sure what the authors were trying to say with the last results section. It is well established that cancers derived from different tissues will have different mutation signatures, often based on their carcinogen exposure or other context specific factors. For example, the authors cite UV signatures in skin cancer, or APOBEC signatures in breast cancer. I was unclear what aspect was being rediscovered here in regards to R-loops? My understanding is that TRCs/R-loops are associated with signatures that are enriched in cancers from tissues where those signatures are themselves enriched. This does not seem to add much new insight as written since it is a circular argument. Perhaps I am missing something but the role of R-loops in this cancer type specificity seems potentially secondary to the real drivers of mutagenesis (e.g. UV exposure).

Reviewer #2 (Remarks to the Author):

In the manuscript, NCOMMS-23-04891-T, Aleix Bayona-Feliu et al studied the regulation of TRC-caused genome instability by various protein factors and histone modifications. In the first part of the paper, the authors described their bench-work results observed from siRNA-mediated depletion of four chromatin regulators, NO80, SNF2H, MTA2 and RB1. The authors then extended their work in the second part of the paper, by integrating their previous DRIPc-seq datasets with public available chip-seq and somatic mutation datasets. However, as shown below, the quality of some of their data needs to be improved and some data interpretation seems doubtful.

Specific concerns:

1. With respect to Figure 1, there are three questions,
 - a. What is the p-value cut off for defining statistical significance? In Fig 1a, the p value between siRB1 and siC is 0.2741, and the authors said the difference is NOT significant. However, in Fig 1c and 1d, the authors said the differences with $p=0.1845$ (FoxP4), $p=0.1746$ (28s rDNA), $p=0.4427$ (5' rDNA), and $p=0.1575$ (5' rDNA-siNO80) are all significant.
 - b. are siMTA2 ($p=0.079$) and Foxp4 ($p=0.0685$) significant?
 - c. Why TAF9B was chosen for siMARCA5, while RPL13A for siNO80?
2. Regarding Extended Data (ED)-Fig 1c, 1d and 1e, it's hard to see the difference between RNH1 + and RNH1 -. It's also hard to see that there were more than 5 rH2AX foci in the siMARCA5, siNO80, siMTA2 and siRB1 images.
3. Regarding Figure 2,
 - a. Figure 2a showed that not just at G2/M, the increase at G1 and S are also significant.
 - b. Again, in figure 2b, what is the p value cutoff for significance?
4. With respect to the enrichment analysis of H3S10,
 - a. The H3S10pho profiles were in the neuroblastoma cell line IMR-5, whereas the DRIPc-seq datasets were produced in K562 cells. Figure 4a and 4b showed that there are significant differences between the genomic profiles of the two features. Has the H3S10pho data normalized?
 - b. In figure 4g, it's unclear how the Hi-C metaplot analysis was conducted, specifically, how the chromatin interaction frequency is normalized and quantified?
5. English language arts need to be improved. There are vague sentences throughout the main text.

Reviewer #3 (Remarks to the Author):

Comments to authors:

Transcription-replication conflicts (TRCs) are known to contribute significantly to genome instability, which can lead to tumorigenesis and cancer progression. *Bayona-Feliu et al* investigated the relationship between TRCs, R-loops, and chromatin regulation by knocking down components of chromatin remodelers, such as SNF2H and INO80, and chromatin-modulating factors, MTA2 and RB1. The authors observed an accumulation of γ H2AX foci in the S/G2 phase in HeLa cells and an increased level of R-loops when these factors were silenced. Additionally, the study revealed multiple factors enriched at TRCs, such as R-loop metabolism, DNA damage response (DDR) components, and histone modifications, while MCM was dis-associated from TRCs. Metadata analysis showed a positive correlation between cancer cell mutations, R loops at head-on TRCs, deficiency in DDR, and chromatin regulators. The authors concluded that the chromatin network plays a role in preventing cancer-associated mutagenesis at TRCs. Although this study provides interesting insights into the role of TRCs and chromatin regulation in cancer genome instability, most of the conclusions are based on correlation analysis, and therefore, more stringent experiments and analysis are needed to confirm these findings.

Major issues:

1. The accumulation of γ H2AX foci represents increased DNA damage and was sensitive to RNH1 treatment, but this alone is not sufficient to establish a correlation with R loops. To strengthen the conclusion that increased double-strand breaks (DSBs) result from R loops, the authors need to map γ H2AX foci or use assays like END-seq to identify genome-wide breaks and compare them with S9.6-enriched sites.
2. For Figure 2c, it would be helpful if the authors could provide a zoomed-in view of the identified peaks for DRIPc, YY1, FANCD2, and SMARCA4/5 to improve clarity. The current view appears to show the signal regions for FANCD2 ChIP-seq as too wide to be called as peaks. Additionally, it should be emphasized in the text or figure legend that the RFD orientation of OK-seq differs from that of the original paper by Petryk et al. (PMID: 26751768).
3. It is important to include detailed methods and analysis in the Results or Methods section to enable evaluation of the validity and reliability of the results. To facilitate understanding, the authors may also consider adding an illustration of the control loci used for ChIP-seq analysis in Fig. 3a. This would help readers to follow the analysis more easily. Moreover, the

authors should ensure that there are no discrepancies between the text and the Methods section regarding the control loci of the CHIP-seq analysis. Such accuracy and clarity are crucial for proper interpretation of the study's findings.

4. The authors used OK-seq and transcription to determine TRC sites. However, it is possible that the observed enrichment of RNA-DNA hybrids (R loops) may be due to the *de novo* synthesis of R loops at DNA breaks in transcribed genes. Previous studies have shown that double-stranded DNA breaks promote the formation of RNA-DNA hybrids (Ohle et al., 2016; PMID: 27881299; Liu et al., 2021; PMID: 33626331). Therefore, the authors should consider discussing this alternative explanation in the manuscript.

Specific points:

1. There are too many abbreviations used in this manuscript, which can make it difficult for readers to follow. Two-letter abbreviations that are frequently used, such as RF, CD, and HO, should be avoided in the text. Additionally, TC-NER in the abstract and CD were not defined.
2. Line 344, figure 5c should be cited.
3. Line 770, colors are mislabeled in the legend.
4. Lines 780, 782, 784, and 785, “ornage” should be “orange”.
5. Legends of Fig. 5b and 5c are mislabeled.

POINT-BY-POINT RESPONSE TO THE REVIEWERS

REVIEWER COMMENTS

Reviewer #1 (Remarks to the Author):

The work by Bayona-Feliu presents a really interesting analysis of mostly published datasets and ENCODE data that is focused understanding correlations of chromatin state determinants with sites of transcription-replication conflict (TRC). This builds on the authors previous work linking the BAF complex to these TRCs. The work is combined with some preliminary analysis of siRNA knockdowns of chromatin remodelers that implicate a few additional players in preventing R-loops associated with TRCs. I felt these two data sets did not go together very well. Notwithstanding the connection to chromatin, the experiments in Figure 1 and 2 feel like part of an incomplete story on INO80 and SMARCA5 regulating R-loops. The data analyses in Figure 3-6 hardly touch on specific chromatin remodelers at all, but rather analyze TRC features to show these are sites of mutagenesis and specific mutation signatures in various types of cancer. So the work is interesting but neither dataset seem complete and they are not well connected to one another. Below are some specific comments, but the major revision really needs to be either assessing roles of INO80/SMARCA5 in a story about that, or analysis of TRC features showing they are mutational hotspots in cancer (both potentially really great stories that are not given a full treatment in this manuscript).

Thanks for this assessment and constructive comments. We value your appreciations and agree that the manuscript would benefit from further cohesion and connection between sections. We rearranged the manuscript sections and added additional data focusing the research on INO80 and SMARCA5, as suggested. We believe that the manuscript can be read now as a single full story on the analysis of the impact of chromatin factors on TRC-associated DNA damage and mutagenesis in cancer, using INO80 and SMARCA5 as models.

1. The introduction states that R-loops are in most cases a threat to genome integrity. Can the authors defend this statement? The roles of R-loops as normal chromatin marks seem to outnumber the roles in genome integrity. R-loops at pause sites, terminators, and other

regions seem to have normal functions. I guess they could have BOTH normal and toxic functions.

Certainly, R-loops have physiological roles; it is not our intention to ignore this. Nevertheless, the presence of such transcription-associated structures hinders DNA replication progression, as shown by many labs. Even though the cell machinery would handle properly physiological R loops, there is sufficient evidence that persistent R-loops pose a threat to genome integrity, in most cases due to the increased transcription-replication conflicts hazardous scenarios. This is the focus of our manuscript. Nevertheless, we agree the message could be confusing and have moderated the sentence. We have changed “are in most cases” by “may pose”.

2. Related to Figure 1 how did the authors decide on these 4 chromatin targeting genes? Did they emerge from a screen or database as candidates? Or from the analyses later in the paper? There are dozens of chromatin binding and regulating factors, especially if we go into histone PTMs. The rationale for this small set of direct tests is poorly justified. Perhaps beginning with the ENCODE analysis would better justify validation of some of these chromatin remodelers.

Thanks for the observation. Yes, we rearranged manuscript sections so they follow a more logical order. Now, the manuscript starts with the bioinformatics screening and then we focus on certain chromatin factors. We also improved the reasoning behind choosing these factors in the text. Thanks for the suggestion.

2. Why do the authors think they see RNaseH1 sensitive DNA damage and S9.6 signal but not FANCD2 foci for INO80 depleted cells? If the story of the manuscript were to focus on INO80 or SMARCA5, additional analyses would be necessary that more directly measuring TRCs, replication dynamics or other features of genome instability that connect the data in Figure 1 to the story.

We apologize and certainly agree that this part was misleading. We have addressed this part experimentally and added more replicates to the FANCD2 analysis of siINO80 cells to prevent confusing results. As can be observed in the new Figure 4b, siINO80 also results in a 2-fold accumulation of FANCD2 foci, similar to SMARCA5. In addition, we have also

measured and quantified TRCs directly by Proximity Ligation Assay (PLA) using antibodies against RNAPII-S2P and PCNA, as suggested. As can be seen in Figure 4a, PLA foci increase significantly in both siNO80 and siSMARCA5 and this phenotype is due to unscheduled R-loops given that such increments are significantly suppressed by RNH1 overexpression. New results are properly discussed in the text.

4. For figure 5 the authors note an enrichment of mutations in R-loop prone regions. As written it was not clear what the comparator data was? In other words, can the authors include clear analyses comparing genes with similar transcription frequencies and other variables that are NOT R-loop prone to make a statement about causality. Or do the features of R-loops (e.g. GC rich, highly transcribed) themselves predict mutation predisposition? It is important to determine whether the observations are truly R-loop specific.

Thanks for this interesting addition. Our data clearly showed sharp mutation enrichments at R-loop-prone regions compared to the surrounding genome regions (Figure 5a-d). However, we agree that little attention was paid to other genome features. To solve this problem and further support the R-loop-specificity of the results we have performed several additional analysis. First, we measured the mutation burden at R-loop regions and compared it with the mutation load of the same regions being randomized along the same genes. Results showed a significant and specific increase in mutation burden at R-loop regions. Second, we performed a metagene mutation analysis on R-loop-prone genes which unveiled mutation patterns closely related to R-loop profiles (Figure 5f). Finally, we determined the mutation hotspots genome-wide by analyzing the data with the MACS2 algorithm, computational tool designed for the detection of enriched areas, and crossed them with genes, differentiating between silenced and expressed genes and whether genes were R-loop-prone or not (Figure 5e). Strikingly, >50% of mutation hotspots correspond to expressed genes harboring R-loops, while only 25.7 and 12.3% colocalize with expressed genes reluctant to R-loop accumulation or silenced genes, respectively. Altogether these data further support the connection between R-loops and gene mutation predisposition.

5. I was not really sure what the authors were trying to say with the last results section. It is well established that cancers derived from different tissues will have different mutation signatures, often based on their carcinogen exposure or other context specific factors. For

example, the authors cite UV signatures in skin cancer, or APOBEC signatures in breast cancer. I was unclear what aspect was being rediscovered here in regards to R-loops? My understanding is that TRCs/R-loops are associated with signatures that are enriched in cancers from tissues where those signatures are themselves enriched. This does not seem to add much new insight as written since it is a circular argument. Perhaps I am missing something but the role of R-loops in this cancer type specificity seems potentially secondary to the real drivers of mutagenesis (e.g. UV exposure).

Thanks for the recommendation. We agree that this section was not adding novelty to the manuscript. As a consequence, we decided to remove this part from the revised manuscript and figures associated.

Reviewer #2 (Remarks to the Author):

In the manuscript, NCOMMS-23-04891-T, Aleix Bayona-Feliu et al studied the regulation of TRC-caused genome instability by various protein factors and histone modifications. In the first part of the paper, the authors described their bench-work results observed from siRNA-mediated depletion of four chromatin regulators, NO80, SNF2H, MTA2 and RB1. The authors then extended their work in the second part of the paper, by integrating their previous DRIPc-seq datasets with public available chip-seq and somatic mutation datasets. However, as shown below, the quality of some of their data needs to be improved and some data interpretation seems doubtful.

Thanks for the comments and suggestions to improve the manuscript. We do think the new version of the manuscript improved significantly. We would also like to apologize if data quality was felt in some cases misleading. We have fixed this in the revised version.

Specific concerns:

1. With respect to Figure 1, there are three questions,
 - a. What is the p-value cut off for defining statistical significance? In Fig 1a, the p value between siRB1 and siC is 0.2741, and the authors said the difference is NOT significant. However, in Fig 1c and 1d, the authors said the differences with $p=0.1845$ (FoxP4),

p=0.1746 (28s rDNA), p=0.4427 (5' rDNA), and p=0.1575 (5' rDNA-siNO80) are all significant.

We apologize for that. This was a mistake. The p-value for statistical significance is set to 0.05, as described in methods sections. We have corrected the text and now only those changes with p-values<0.05 are treated as significant. Also, we have performed additional experiments and added this new data to reach significant differences (p-value<0.05) when possible.

b. are siMTA2 (p=0.079) and Foxp4 (p=0.0685) significant?

We apologize again. As mentioned before, the text has been corrected and we have experimentally addressed these issues in order to achieve significant results when possible.

c. Why TAF9B was chosen for siMARCA5, while RPL13A for siNO80?

Thanks for the comment. We have now homogenized the analysis by adding additional experimental data so that the same genes are analyzed in both conditions (siSMARCA5 and siNO80).

2. Regarding Extended Data (ED)-Fig 1c, 1d and 1e, it's hard to see the difference between RNH1 + and RNH1 -. It's also hard to see that there were more than 5 rH2AX foci in the siSMARCA5, siNO80, siMTA2 and siRB1 images.

We appreciate this observation. In addition to wide-field images shown in Extended Data Figure 3 in the revised manuscript, we have also added clear single-cell example images in the main figures so that foci and the differences can be appreciated (Figure 3a,b; Figure 4a,b).

3. Regarding Figure 2,

a. Figure 2a showed that not just at G2/M, the increase at G1 and S are also significant.

Yes, there is a significant increase in siSMARCA5 in G1, but milder than the ones occurring in S or G2/M. This increase in G1 might be the result of DNA damage being carried from previous S phases of the cell cycle. Nevertheless, the objective of this analysis was to detect a major accumulation of DNA damage during S+G2/M consistent with replication impairment and data clearly show that the main accumulation of DNA breaks occurs during S+G2/M, which is where we have focused the study.

b. Again, in figure 2b, what is the p value cutoff for significance?

As mentioned above, we have added new experimental data and addressed this question in the text.

4. With respect to the enrichment analysis of H3S10,

a. The H3S10pho profiles were in the neuroblastoma cell line IMR-5, whereas the DRIPc-seq datasets were produced in K562 cells. Figure 4a and 4b showed that there are significant differences between the genomic profiles of the two features. Has the H3S10pho data normalized?

Thanks for the observation, this part was missing in the methods section. We have now fixed it. H3S10pho data were RPKM normalized to obtain coverage files.

b. In figure 4g, it's unclear how the Hi-C metaplot analysis was conducted, specifically, how the chromatin interaction frequency is normalized and quantified?

We apologize if the HiC analysis description was misleading, we have now added a detailed description of the methodology used in the methods section.

5. English language arts need to be improved. There are vague sentences throughout the main text.

We apologize, we have now improved manuscript cohesion and text quality in the revised version.

Reviewer #3 (Remarks to the Author):

Comments to authors:

Transcription-replication conflicts (TRCs) are known to contribute significantly to genome instability, which can lead to tumorigenesis and cancer progression. Bayona-Feliu et al investigated the relationship between TRCs, R-loops, and chromatin regulation by knocking down components of chromatin remodelers, such as SNF2H and INO80, and chromatin-modulating factors, MTA2 and RB1. The authors observed an accumulation of γ H2AX foci in the S/G2 phase in HeLa cells and an increased level of R-loops when these factors were silenced. Additionally, the study revealed multiple factors enriched at TRCs, such as R-loop metabolism, DNA damage response (DDR) components, and histone modifications, while

MCM was dis-associated from TRCs. Metadata analysis showed a positive correlation between cancer cell mutations, R loops at head-on TRCs, deficiency in DDR, and chromatin regulators. The authors concluded that the chromatin network plays a role in preventing cancer-associated mutagenesis at TRCs. Although this study provides interesting insights into the role of TRCs and chromatin regulation in cancer genome instability, most of the conclusions are based on correlation analysis, and therefore, more stringent experiments and analysis are needed to confirm these findings.

Thanks for the constructive comments and recommendations. Indeed, we agree additional experimental data would further support our results. Thus, we performed PLA using antibodies against RNAPII-S2P and PCNA to directly confirm TRC enrichments in siSMARCA5 and siINO80 cells. Furthermore, we also analyzed mutagenesis in tumors deficient of SMARCA4, SMARCA5 and INO80 to further confirm the association between malfunctioning of these enzymes and increase mutagenesis at R-loops in cancer. Finally, we have also added additional experimental replicates when needed to the presented data to achieve significant and clearer results, as suggested. Details are indicated below. Thanks for making us improve the manuscript.

Major issues:

1. The accumulation of γ H2AX foci represents increased DNA damage and was sensitive to RNH1 treatment, but this alone is not sufficient to establish a correlation with R loops. To strengthen the conclusion that increased double-strand breaks (DSBs) result from R loops, the authors need to map γ H2AX foci or use assays like END-seq to identify genome-wide breaks and compare them with S9.6-enriched sites.

Thanks for the comment and suggestion. However, first we would like to stress that our data confirm sufficiently that hybrids induce the breaks. RNH1 is a ribonuclease that degrades the RNA moiety of the DNA-RNA hybrid widely used as a control of R-loop-dependent phenotypes, as shown by the lab of Bob Crouch at NIH (Cerritelli, S.M. and Crouch, R.J. 2009, 2014) and many others. The fact that our observed phenotypes decrease significantly upon overexpression of the enzyme confirm that the R-loops mediate the observed phenotype. In addition, strong correlation between R-loops and DNA damage was shown in many previous publications, including our recent on the SWI/SNF factor (Bayona-Feliu, A. et al. 2021). A genome-wide analysis like the proposed one is certainly interesting and will get into fine-mapping of breaks, but does not warrant to obtain the information required because DRIP-seq is not sufficiently resolutive as to obtain data comparable to END-seq.

In addition, a priori, it is not possible to predict whether breaks occur at the hybrid site itself or besides, provided that if in most cases breaks are associated with replication fork stalls, we do not know where these take place and break. The suggested experiments are certainly important, but they are a project by itself that would need much more additional data and it is beyond the scope of this manuscript.

2. For Figure 2c, it would be helpful if the authors could provide a zoomed-in view of the identified peaks for DRIPc, YY1, FANCD2, and SMARCA4/5 to improve clarity. The current view appears to show the signal regions for FANCD2 ChIP-seq as too wide to be called as peaks. Additionally, it should be emphasized in the text or figure legend that the RFD orientation of OK-seq differs from that of the original paper by Petryk et al. (PMID: 26751768).

Thanks for the suggestion, we agree and added zoomed-in snapshots where correlations can be clearly appreciated. We also fixed the issue regarding RFD.

3. It is important to include detailed methods and analysis in the Results or Methods section to enable evaluation of the validity and reliability of the results. To facilitate understanding, the authors may also consider adding an illustration of the control loci used for ChIP-seq analysis in Fig. 3a. This would help readers to follow the analysis more easily. Moreover, the authors should ensure that there are no discrepancies between the text and the Methods section regarding the control loci of the ChIP-seq analysis. Such accuracy and clarity are crucial for proper interpretation of the study's findings.

We apologize if this was not clear. We added new elements to the scheme presented in Figure 3a (now, Figure 1a) and improved methodology description in the Methods section to facilitate understanding of our analysis.

4. The authors used OK-seq and transcription to determine TRC sites. However, it is possible that the observed enrichment of RNA-DNA hybrids (R loops) may be due to the de novo synthesis of R loops at DNA breaks in transcribed genes. Previous studies have shown that double-stranded DNA breaks promote the formation of RNA-DNA hybrids

(Ohle et al., 2016; PMID: 27881299; Liu et al., 2021; PMID: 33626331). Therefore, the authors should consider discussing this alternative explanation in the manuscript.

Thanks for the comment. We are aware of the induction of hybrids by DSBs, which we have reviewed and discussed in *Nat Struct Mol Biol* (Gómez-González and Aguilera 2017), findings first reported by Lieber (Roy et al, *MCB* 2010) and Calsou (Britton et al, *NAR* 2014). As indicated before the fact that RNH1 suppresses DSBs supports the view that in our case the hybrid precedes the DSB and not the vice versa. However, we agree that we should not disregard this possibility and, following the suggestions, we have added a sentence considering the possibility that some hybrids may be induced by previously formed DSBs. Having said that, this is a question that could be applied to any other study showing increase in hybrids that does not change at all the main conclusion, and one major reason why it is essential to suppress R-loop-mediated damage by RNH1, as we did.

Specific points:

1. There are too many abbreviations used in this manuscript, which can make it difficult for readers to follow. Two-letter abbreviations that are frequently used, such as RF, CD, and HO, should be avoided in the text. Additionally, TC-NER in the abstract and CD were not defined.

Thanks, we understand the point. These abbreviations are frequently used in the field and help shorten the sentences to make them more readable. In any case, we have followed the suggestion of removing the two-letter abbreviations RF, CD and HO, and have defined the meaning of TC-NER. Thanks.

2. Line 344, figure 5c should be cited. **Thanks, fixed.**

3. Line 770, colors are mislabeled in the legend. **Thanks, fixed.**

4. Lines 780, 782, 784, and 785, “ornage” should be “orange”. **Thanks, fixed.**

5. Legends of Fig. 5b and 5c are mislabeled. **Thanks, fixed.**

REVIEWERS' COMMENTS

Reviewer #1 (Remarks to the Author):

The authors have provided a thoughtful and thorough revision. The reorganized manuscript flows much better and the inclusion of new data, replicates, and statistics helps to further support their conclusions. The work really expands on the connections of R-loop and TRC biology to chromatin state. I have no further suggested revisions.

Reviewer #2 (Remarks to the Author):

The revised manuscript has addressed all my concerns raised in the previous round of review. I have no further concerns.

Reviewer #3 (Remarks to the Author):

I am satisfied the authors have addressed most of the questions, but some minor concerns remained to be addressed before publication.

1. Lots of p values have been changed, the authors need to present and publish all the original data for the changed figure panels to rational these changes.
2. In lines 335-338, what are the percentages of R-loop regions overlapped with SMARCA4, SMARCA5 and YY1 peaks (not target genes)? Are they significant or not?
3. In lines 400-402 and Fig 5f, what are the percentages of R-loop-prone genes? Whether the density of mutations on R-loop-prone genes is significantly higher than R-loop-reluctant and silenced genes? The density is a better way to compare.

POINT-BY-POINT RESPONSE TO THE REVIEWERS

REVIEWERS' COMMENTS

Reviewer #1 (Remarks to the Author):

The authors have provided a thoughtful and thorough revision. The reorganized manuscript flows much better and the inclusion of new data, replicates, and statistics helps to further support their conclusions. The work really expands on the connections of R-loop and TRC biology to chromatin state. I have no further suggested revisions.

Thanks for the constructive review and for the suggestions that contributed to improve manuscript cohesion and strengthen the paper main conclusions.

Reviewer #2 (Remarks to the Author):

The revised manuscript has addressed all my concerns raised in the previous round of review. I have no further concerns.

Thanks for the assessment of the manuscript and the suggestions that helped improve the manuscript.

Reviewer #3 (Remarks to the Author):

I am satisfied the authors have addressed most of the questions, but some minor concerns remained to be addressed before publication.

Thanks for the comments and suggestions, they enhanced manuscript quality. We apologize if these minor concerns were unclear.

1. Lots of p values have been changed, the authors need to present and publish all the original data for the changed figure panels to rational these changes.

Certainly, P-values of some conditions slightly changed from previous panels because we added new data from the experiments required during the review process. Therefore, addition of new biological replicates slightly modified the P-values in certain cases. Furthermore, we uploaded all source data for the review process, as requested.

2. In lines 335-338, what are the percentages of R-loop regions overlapped with SMARCA4, SMARCA5 and YY1 peaks (not target genes)? Are they significant or not?

We compared target genes instead of peaks because no data support an R-loop resolvase activity for chromatin remodelers, as it is the case of DNA-RNA helicases like DDX39b/UAP56 (Perez-Calero, C. et al. Genes Dev, 2020). Indeed, the chromatin remodeling might not be required at the same exact genome position of R-loop formation, as current data support the view that a main source of R-loop-induced DNA damage is through replication fork stalling, as outlined in the manuscript introduction. Therefore, comparisons at peak level are less informative than at gene level in this case. For these reasons, we think that measuring the overlap between chromatin remodelers and R-loops over genes is more appropriate in our case.

3. In lines 400-402 and Fig 5f, what are the percentages of R-loop-prone genes? Whether the density of mutations on R-loop-prone genes is significantly higher than R-loop-reluctant and silenced genes? The density is a better way to compare.

This is an interesting question, thanks for the observation. We have re-analyzed the data and compared the expected number of mutation hotspots at the different gene types analyzed based on the genome frequency of each gene type to avoid any possible bias with the observed values, as suggested. Consistently, we found a significant enrichment of mutation hotspots at R-loop-prone genes, which further support the view that mutagenesis is enhanced at R-loop-forming genes in cancer. The new data are in the new Supplementary Figure 5c and discussed in the text. Thanks for this addition.